# Purkinje cell neurotransmission patterns cerebellar basket cells into zonal modules defined by distinct pinceau sizes

Joy Zhou[1,2,3†], Amanda M Brown[1,2,3†], Elizabeth P Lackey[1,2,3], Marife Arancillo[1,3], Tao Lin[1,3], Roy V Sillitoe[1,2,3,4*]

[1]Department of Pathology and Immunology, Baylor College of Medicine, Houston, United States; [2]Department of Neuroscience, Baylor College of Medicine, Houston, United States; [3]Jan and Dan Duncan Neurological Research Institute of Texas Children's Hospital, Houston, United States; [4]Program in Developmental Biology, Baylor College of Medicine, Houston, United States

**Abstract** Ramón y Cajal proclaimed the neuron doctrine based on circuit features he exemplified using cerebellar basket cell projections. Basket cells form dense inhibitory plexuses that wrap Purkinje cell somata and terminate as pinceaux at the initial segment of axons. Here, we demonstrate that HCN1, Kv1.1, PSD95 and GAD67 unexpectedly mark patterns of basket cell pinceaux that map onto Purkinje cell functional zones. Using cell-specific genetic tracing with an *Ascl1^CreERT2* mouse conditional allele, we reveal that basket cell zones comprise different sizes of pinceaux. We tested whether Purkinje cells instruct the assembly of inhibitory projections into zones, as they do for excitatory afferents. Genetically silencing Purkinje cell neurotransmission blocks the formation of sharp Purkinje cell zones and disrupts excitatory axon patterning. The distribution of pinceaux into size-specific zones is eliminated without Purkinje cell GABAergic output. Our data uncover the cellular and molecular diversity of a foundational synapse that revolutionized neuroscience.

*For correspondence:
sillitoe@bcm.edu

†These authors contributed equally to this work

## Introduction

Studies of the cerebellar basket cell, first by Camillo Golgi and then by Santiago Ramón y Cajal, hold a special place in history. In particular, it was Cajal's discovery that the endings of basket cells terminate upon what would become known as the initial segment of Purkinje cells that sparked a new era of neuroscience (*Cajal, 1911*). He called this nerve ending 'the pinceau', named for its paintbrush-like appearance. Anatomical analyses revealed the complexity of this synapse as a dense and intriguing set of contacts that played a key role in the debate of whether neurons were individual units connected by synapses, or whether they were unified in a reticulum with a somewhat uninterrupted flow of information. The complexity of the basket cell pinceau hid its true connectivity when studied using the Golgi reaction, although using electron microscopy, Sanford Palay and Victoria Chan-Palay resolved the full architecture of the basket cell axons, their collaterals, the pericellular baskets that wrap around the Purkinje cell soma, and the pinceau terminals that contact the initial segment of the Purkinje cell axon (*Palay and Chan-Palay, 1974*). The surprising sparseness of synaptic contacts between the pinceau and the Purkinje cell axon (*Palay and Chan-Palay, 1974*; *Somogyi and Hámori, 1976*) – although reliably found on the Purkinje cell axons of different species (*Hámori and Szentágothai, 1965*) – was, at the time, consistent with the relatively weak functional inhibitory connectivity shown by slice electrophysiology recordings (*Roberts, 1968*; *Korn and Axelrad, 1980*). More than three decades later, advanced slice electrophysiology recording methods revealed an unexpected ultra-fast ephaptic mode of axon-to-axon communication between basket cells and

Purkinje cells (*Blot and Barbour, 2014*). Accordingly, the collective repertoire of contacts between the two cell types makes a substantial functional contribution, as genetic silencing of GABAergic basket cell output alters Purkinje cell firing in vivo (*Brown et al., 2019*). There is also evidence showing that basket cells play an essential role in controlling cerebellar cortical output during motor behavior (*Barmack and Yakhnitsa, 2008*; *Wulff et al., 2009*; *Jelitai et al., 2016*; *Gaffield and Christie, 2017*; *He et al., 2015*; *Sergaki et al., 2017*). Interestingly, basket cells project in the sagittal plane (*Palay and Chan-Palay, 1974*), which is intriguing because Purkinje cell molecular and functional heterogeneity are restricted to sagittal domains (*Apps et al., 2018*). Here, we investigated basket cell connectivity based on how the pericellular baskets and pinceau terminals, in particular, are connected within Purkinje cell sagittal maps (*Miterko et al., 2018*). This missing information is crucial for understanding how basket cells communicate with Purkinje cells, especially since the basket cells are coupled in sagittal rows (*Sotelo, 2015*). The electrical and chemical connectivity coefficients of basket cells are strongly represented in the sagittal plane (*Rieubland et al., 2014*). However, it is unclear how this functional organization fits into that of the broader cerebellar map with its complex but systematic patterns of topographic connectivity (*Apps et al., 2018*).

Cerebellar circuit maps are comprised of hundreds, perhaps thousands, of modules (*Apps et al., 2018*; *Miterko et al., 2018*). Each module is assembled from an array of cell types that are arranged around Purkinje cell patterns (*Sillitoe and Joyner, 2007*; *Apps and Hawkes, 2009*). The surrounding cells are all patterned and include excitatory granule cells and unipolar brush cells (*Sillitoe et al., 2003*; *Chung et al., 2009*; *Lee et al., 2015*), inhibitory Golgi cells (*Sillitoe et al., 2008a*), and even Bergmann glia (*Reeber et al., 2018*). Excitatory climbing fiber and mossy fiber afferents also terminate in domains that respect Purkinje cell zones (climbing fibers–*Gravel et al., 1987*; *Sugihara and Shinoda, 2007b*; *Reeber and Sillitoe, 2011*; mossy fibers–*Brochu et al., 1990*; *Quy et al., 2011*; *Gebre et al., 2012*). In this study, we address whether molecular layer (ML) inhibitory interneurons are also patterned into zones. We use conditional genetic labeling and neuronal silencing in mice to uncover a size-based segregation of basket cell projections into zones. We reveal that basket cell pinceaux have different sizes, and their sizes are determined cell non-autonomously by Purkinje cell GABAergic neurotransmission. These data are critical for establishing a complete in vivo model for how the cerebellum functions during motor and cognitive tasks.

## Results

### Cerebellar basket cell interneurons have a complex structural interaction with Purkinje cells

The cerebellar nuclei mediate the motor and non-motor functions of the cerebellum using ascending and descending projections to the thalamus, red nucleus, and inferior olive (*Figure 1A*). However, before the information is communicated out of the cerebellum, it is processed in the cerebellar cortex by a relatively small number of excitatory and inhibitory neuron classes (*Figure 1A*). The cerebellar cortex has three distinct layers (*Figure 1A*). The most superficial layer contains inhibitory interneurons called basket cells and stellate cells, as well as excitatory climbing fibers and parallel fibers (the axons of granule cells). All four cell types project onto the Purkinje cells, which make up the middle cerebellar cortical layer called the Purkinje cell layer (PCL). The PCL also contains candelabrum cells and large astrocytes called Bergmann glia. Purkinje cells perform the main computations in the cerebellum. The deepest layer is called the granular layer (GL), and contains millions of excitatory neurons called granule cells, a smaller population of excitatory neurons called unipolar brush cells, inhibitory Lugaro cells, and input fibers called mossy fibers that deliver sensory signals to the cerebellum from dozens of brain and spinal cord nuclei (*Figure 1A*; *White and Sillitoe, 2013*). The interactions between cerebellar cortical neurons depend on their individual cellular structures as well as their patterning in the coronal and sagittal planes. Here, we focus on the underappreciated architecture, patterning, and connectivity of the basket cells (*Figure 1B*).

Staining using a modified version of the Golgi-Cox method reveals the dense axonal projections of the basket cells around Purkinje cells (*Figure 1B*). The descending branches of basket cell axons enwrap the cell body of Purkinje cells, making perisomatic synapses, but they also extend to reach the axon initial segment (AIS). A remarkable feature of this GABAergic innervation of Purkinje cells is the basket cell pinceau, a peculiar assembly of basket cell axons around the AIS of Purkinje cells

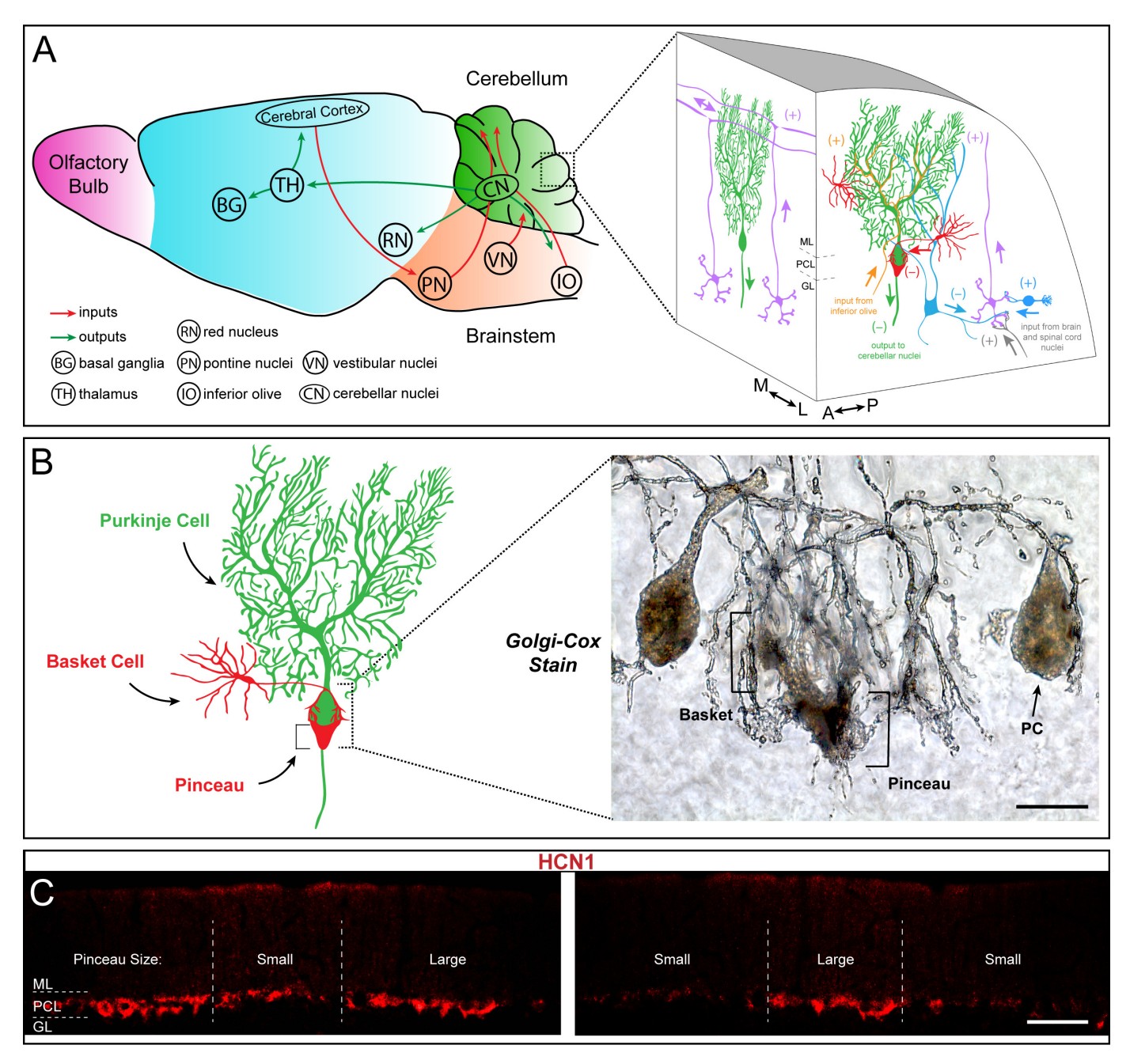

**Figure 1.** Basket cells are inhibitory interneurons in the cerebellar cortex that innervate Purkinje cells with a unique terminal called the pinceau. (**A**) Left: Schematic of sagittal tissue section through the mouse brain illustrating key inputs (red arrows) and outputs (green arrows) between the cerebellum and other major brain regions. For reference, general divisions of the brain including the cerebellum (green), brainstem (orange), cerebral cortex (blue), and olfactory bulb (purple) are color coded. Right: Magnified schematic, depicted as a 3-dimensional image, of the cerebellar cortex showing the main cell types including Purkinje cells (green), granule cells (purple), and basket and stellate cells (red). Purkinje cell somata are contained in the Purkinje cell layer (PCL) underneath the molecular layer (ML), and directly below the PCL lies the granular layer (GL) containing granule cells and various classes of interneurons (blue). (+) and (–) indicate excitatory and inhibitory synapses, respectively. Known orientations of projections and cell morphologies are presented in both the sagittal ((A) anterior, (P) posterior) and coronal ((M) medial, (L) lateral) planes. (**B**) Left: Schematic of a Purkinje cell (PC, green) with an innervating basket cell (BC, red). Right: Golgi-Cox staining reveals the intricate innervation of basket cell axons onto the Purkinje cell soma and the axon initial segment (AIS). The ascending collaterals are not easily appreciated here. Basket cell axons initially form branching contacts on the somata of Purkinje cells, creating a basket-like shape (left bracket). Upon reaching the AIS, the axons extend terminal branches that converge to form the pinceau (right bracket). Scale bar is 15 µm. (**C**) Coronal-cut cerebellar tissue sections from an adult mouse stained for HCN1, which reveals the zonal

*Figure 1 continued on next page*

*Figure 1 continued*

patterning of basket cell pinceau projections. Dotted lines indicate zone boundaries. PCL, ML, and GL are indicated by PCL, ML, and GL, respectively. Basket cell pinceaux are located in the most superficial regions of the GL. Scale bar is 100 µm.

(*Cajal, 1911*; *Chan-Palay and Palay, 1970*). With the prediction that the same organization is found in all regions of the cerebellum, we used HCN1 to examine basket cell connectivity in more detail. HCN1, or hyperpolarization-activated cyclic nucleotide-gated potassium channel 1, is a membrane protein that contributes to native pacemaker currents in the heart and nervous system (*Chang et al., 2019*). The four HCN channels are encoded by the *HCN1-4* genes and together they modulate cellular excitability, rhythmic activity, dendritic integration, and synaptic transmission (*Moosmang et al., 1999*; *Moosmang et al., 2001*; *Notomi and Shigemoto, 2004*; *He et al., 2014*). In the cerebellum, HCN1 is expressed in Purkinje cells, where it mediates a large hyperpolarization-activated current ($I_h$) (*Nolan et al., 2003*). However, it is also heavily expressed presynaptically in basket cell terminals (*Santoro et al., 1997*; *Luján et al., 2005*). Unexpectedly, we found that HCN1 shows a non-uniform pattern of expression on tissue sections cut through the adult mouse cerebellum (two different areas of cerebellar cortex are shown in *Figure 1C*). The unequal distribution of HCN1 around the base of Purkinje cells suggested that some basket cells either express more HCN1, or express it at higher intensity, compared to their neighbors. The patchy staining also raised the possibility that presynaptic HCN1 is expressed in a systematic pattern in the cerebellum. We therefore used a combination of marker analyses and genetic manipulations to test these different possibilities.

## HCN1 expression in basket cell terminals respects the zonal patterning of Purkinje cells

The heterogeneous distribution of HCN1 at basket cell terminals hinted at a possible zonal pattern of expression in which some basket cells might express more HCN1 than others, or at the extreme, some express it whereas others do not. Cerebellar zonal patterning is a fundamental architecture that is respected not only by Purkinje cells, but also by their afferent and interneuron microcircuit components (*Apps and Hawkes, 2009*; *Cerminara et al., 2015*). The precision of zonal connectivity provides a structural framework for understanding how circuits operate during ongoing motor function and motor learning (*Attwell et al., 1999*; *Wadiche and Jahr, 2005*; *Horn et al., 2010*; *Mostofi et al., 2010*; *Cerminara and Apps, 2011*; *Graham and Wylie, 2012*). Importantly, the behavioral correlates of zonal circuitry may be determined at the level of cellular firing activity (*Zhou et al., 2014*; *Xiao et al., 2014*), and indeed if Purkinje cell neurotransmission is manipulated, zonal patterning is disrupted (*White et al., 2014*). Based on these data, the growing assumption is that all cerebellar components are zonally patterned, but we have only limited experimental evidence for such organization for certain cell types. Of specific relevance, based on Golgi-Cox staining we have previously demonstrated the possibility that stellate cell interneurons, specifically their somata, are restricted at Purkinje cell zonal boundaries in the ML (*Sillitoe et al., 2008a*). However, based on the randomness of staining using the Golgi-Cox method and the limited ability to track distinct subsets of cells and their respective projections with full clarity, we could not with confidence make any conclusion about how basket cells are organized (*Sillitoe et al., 2008a*). The potential of HCN1 expression to fill this gap in our knowledge motivated a double-staining experiment using HCN1 and zebrinII (*Figure 2*). ZebrinII is a polypeptide antigen found on the aldolase C protein (*Ahn et al., 1994*; *Brochu et al., 1990*). Lobules I-V and anterior VIII-IX are identified by a striking array of zebrinII zones, where lobules VI-VII and posterior IX-X express it uniformly (*Sillitoe and Hawkes, 2002*). We therefore analyzed HCN1 expression in lobule VIII due to the clarity of the individual zones (*Figure 2A*) as defined by the sharpness of zonal boundaries (*Figure 2B*), and because the zones abutting the zebrinII P1+ midline zone in lobule VIII are roughly equal in width; the number of Purkinje cells in a zebrinII-expressing zone is equal to the number of Purkinje cells in an adjacent zone that does not express the antigen (*Brochu et al., 1990*; *Ozol et al., 1999*). We found that the pattern of HCN1 indeed respected the pattern of zebrinII, with an inverse relationship between the two. HCN1 expression was more prominent around Purkinje cells that did not express zebrinII (*Figure 2C–F*), with this relationship best appreciated at zone boundaries where zebrinII non-expressing cells have a robust HCN1 profile compared to the immediately adjacent zebrinII-

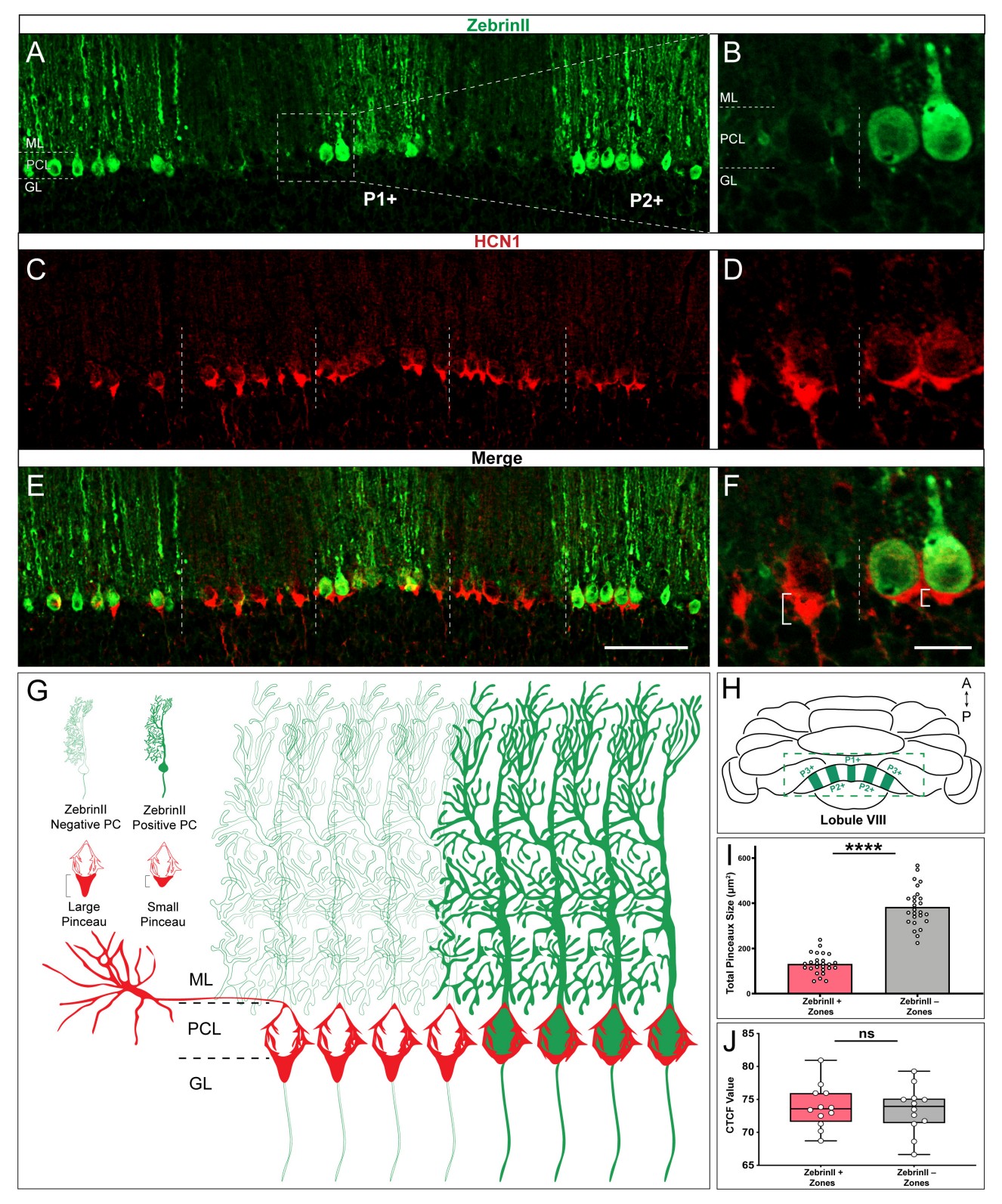

**Figure 2.** HCN1-labeled basket cell pinceaux are smaller in zebrinII-positive zones and larger in zebrinII-negative Purkinje cell zones. (A, C, E) Coronal sections cut through the cerebellar cortex showing zebrinII (green, PC) and HCN1 (red, pinceau) expression. Dotted lines delineate the Purkinje cell zonal boundaries. Purkinje cell bodies are contained within the Purkinje cell layer (PCL) underneath the molecular layer (ML), and basket cell pinceaux are located in the superficial granular layer (GL) and PCL. (B, D, F) Magnified image of a zebrinII zonal boundary from panel A, C, and E, respectively

*Figure 2 continued on next page*

*Figure 2 continued*

(left, zebrinII-negative; right, zebrinII-positive). (E,F) Merged zebrinII and HCN1 expression patterns from A–D. Scale bars are 100 μm and 30 μm, respectively. Brackets in F highlight the pinceau size difference across a zebrinII Purkinje cell zonal boundary. (G) Schematic depiction of pinceau size distinctions in zebrinII-positive and -negative zones. (H) Whole-mount schematic diagram of the cerebellum showing the zebrinII expression pattern in lobule VIII. ZebrinII-positive zones in green are marked as P1+, P2+, and P3+ using the standard zebrinII zone nomenclature (see *Sillitoe and Hawkes, 2002*). (I) Quantification of pinceau area across zebrinII Purkinje cell zones in C57BL/6J mice reveals significantly smaller total pinceau size in zebrinII-positive zones (mean = 131.4 μm$^2$, SD = 44.76 μm$^2$) compared to negative zones (mean = 383.5 μm$^2$, SD = 87.19 μm$^2$). Each data point indicates the total area of multiple HCN1-labeled pinceaux within a 100 μm-wide region of a zebrinII-positive or -negative Purkinje cell zone, reported in μm$^2$ ($N$ = 6, $n$ = 12 sections, 26 zebrinII-positive Purkinje cell zones and 26 zebrinII-negative Purkinje cell zones; ****p<0.0001). (J) Corrected total cell fluorescence (CTCF) analysis reveals no significant difference in HCN1-labeled pinceau fluorescence intensity between pinceaux associated with zebrinII-positive (mean = 73.9, SD = 3.3) and zebrinII-negative (mean = 73.41, SD = 3.55) Purkinje cells. Each data point represents the CTCF value of a 1 μm$^2$ region in a single pinceau ($N$ = 6 mice, $n$ = 12 large and 12 small pinceaux; p>0.05; note, however, that although six mice were used for the quantitative analysis, the patterned relationship between HCN1 and zebrinII was consistently observed in every mouse studied so far, $N$ > 20).

The online version of this article includes the following source data and figure supplement(s) for figure 2:

**Source data 1.** Source data for representative graphs in *Figure 2*.
**Figure supplement 1.** HCN1 expression reveals zones in the hemisphere lobules.
**Figure supplement 2.** Sample antibody staining controls for the expression of protein markers in basket cell zones.

expressing cells that have reduced prominence of HCN1 profiles (*Figure 2D,F*). We next tested whether the HCN1-expressing profiles were different sizes. Specifically, we tested whether there is restricted expression of the protein, or differences in the intensity of expression but within equally sized profiles around Purkinje cells. We quantified pinceau expression in the P1+ to P3+ zones (and intervening P- zones) of lobule VIII (*Figure 2H*) and found a significant difference in the size of pinceaux between zebrinII-positive and zebrinII-negative zones (*Figure 2I*). We then tested whether this size difference was driven by an unequal intensity of protein expression. We found no difference in HCN1 intensity between pinceaux of different sizes (*Figure 2J*). Despite the differences we uncovered in pinceaux size, we also observed some degree of heterogeneity of pinceau size within molecularly defined zones. Pinceau size variance within individual zones of zebrinII-positive or -negative identity can be explained by the general anatomical organization of basket cells within the cerebellum (they are restricted to the vicinity around the monolayer of Purkinje cells, which are impressively arranged but not always perfectly aligned), how the tissue was cut and how it was eventually imaged and visualized for analysis. Since the basket cell pinceau structure is somewhat conical in nature, any section that does not perfectly bisect the point of the pinceau cone will not reflect its maximum height and width. Combined with the fact that each pinceau may naturally lay a few micrometers offset from its neighbor, as these afferents adopt the occasionally imperfect alignment of their target Purkinje cells, each of the 40-micrometer-thick coronal tissue sections we used to visualize pinceau patterning inevitably reveal small differences in pinceau size, even within the same zone. However, to minimize these variances as much as possible, only Z-stack images spanning multiple micrometers of tissue in each section were imaged, analyzed, and displayed. Importantly, our analyses have shown that overall pinceau size is significantly different between zebrinII zones irrespective of the heterogeneity between pinceaux that exist in a given zone. Although we focused our analysis on the vermis, we also observed a similar patterning of HCN1 into parasagittal zones in the hemisphere lobules (*Figure 2—figure supplement 1*). Overall, these data suggested that zebrinII-positive zones were populated with basket cells with small pinceaux, while zebrinII-negative zones were populated with basket cells with large pinceaux (*Figure 2G*).

However, zebrinII is not the only marker of zones (*White and Sillitoe, 2013*). In some cases, zebrinII zones are complementary to the expression pattern of proteins such as phospholipase C β4 (PLCβ4; *Armstrong and Hawkes, 2000*; *Sarna et al., 2006*), while in other cases they are co-expressed with proteins such as phospholipase C β3 (PLCβ3; *Armstrong and Hawkes, 2000*; *Sarna et al., 2006*). We therefore co-stained coronal-cut tissue sections with HCN1 and PLCβ4 (*Figure 3A–F*) and found that indeed, larger HCN1-expressing basket cell profiles localized around PLCβ4-expressing/zebrinII non-expressing Purkinje cells (P- zones in *Figure 3G–H*).

In addition to complementary patterns of expression in lobules with zebrinII zones, there are also markers that label zones within lobules with Purkinje cells that all express zebrinII. Lobules VI-VII and posterior IX-X express the small heat shock protein HSP25 in zones (*Armstrong and Hawkes,*

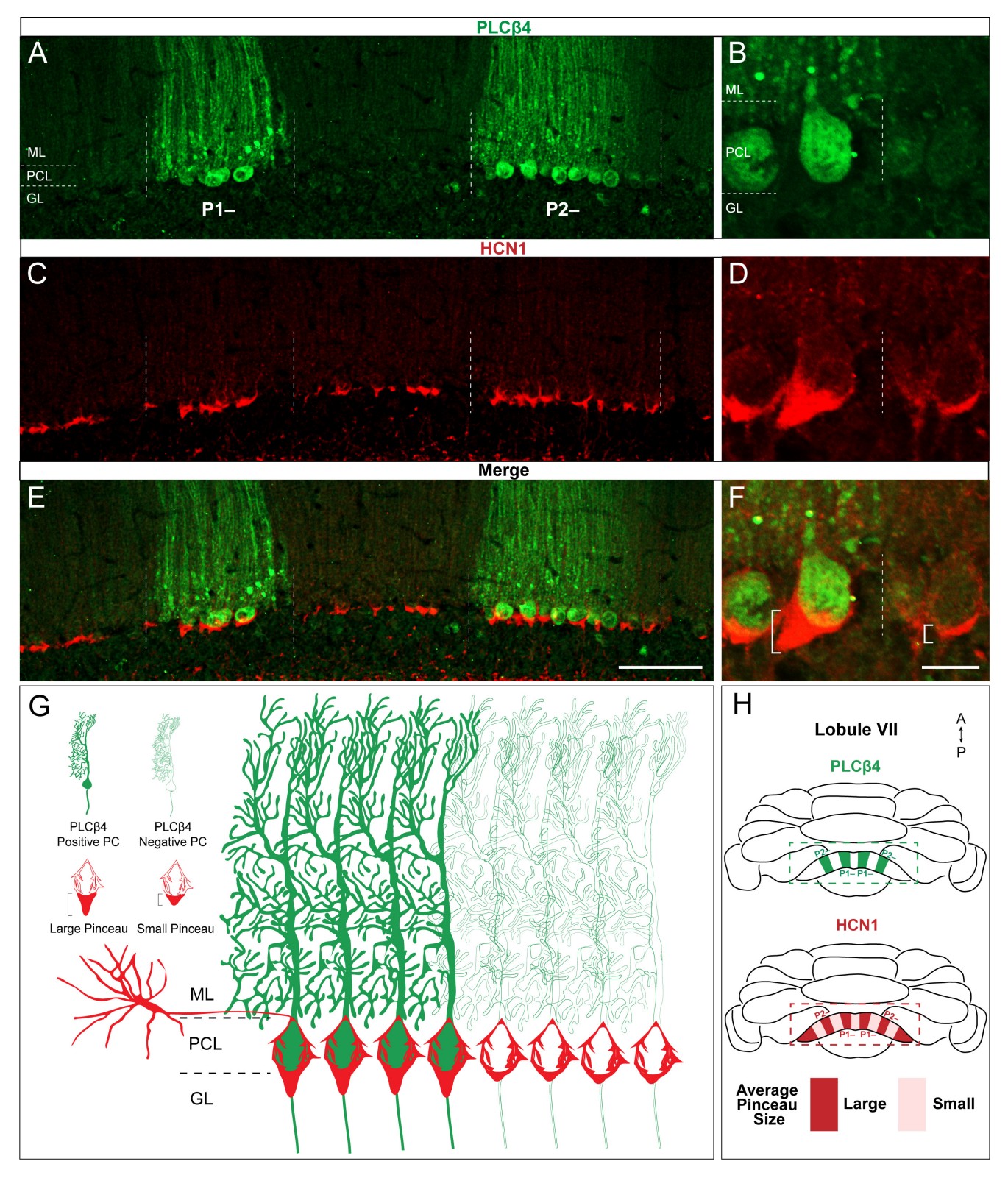

**Figure 3.** HCN1-labeled basket cell pinceaux are larger in PLCβ4-positive Purkinje cell zones. (**A, C, E**) Coronal sections cut through the cerebellar cortex showing PLCβ4 (green, PC) and HCN1 (red, pinceau) expression. Dotted lines indicate Purkinje cell zonal boundaries. Purkinje cell somata are contained within the Purkinje cell layer (PCL) underneath the molecular layer (ML), and basket cell pinceaux are observed in the granular layer (GL) and occasionally in the PCL. (**B, D, F**) Higher magnification view of a PLCβ4 zonal boundary (left, PLCβ4-positive; right, PLCβ4-negative). Scale bars are 100

*Figure 3 continued on next page*

*Figure 3 continued*

μm and 30 μm, respectively. Brackets in (F) highlight the pinceau size difference across a PLCβ4 Purkinje cell zonal boundary. (G) Schematic depiction of the pinceau size differences in PLCβ4-positive and -negative zones. (H) Whole-mount schematic diagram of the cerebellum showing the PLCβ4 and HCN1 expression patterns in lobule VIII. PLCβ4-positive zones in green are marked as P1- and P2- using the standard zebrinII zone nomenclature (*Ozol et al., 1999*; *Sillitoe and Hawkes, 2002*). Differences in HCN1-labeled pinceau sizes across PLCβ4 zones are labeled in dark red and light red, with larger pinceaux (dark red) located on Purkinje cells within the PLCβ4-positive zones (*N* = 4).

*2000*), and we previously showed that the pattern of neurofilament heavy chain (NFH) expression is complementary to HSP25 in these specific lobules (*Demilly et al., 2011*). In addition, because NFH reveals zones across multiple sets of lobules (*Demilly et al., 2011*; *White and Sillitoe, 2013*) and because the robustness of NFH within both the Purkinje cells and the 'basket', or somata portion of the basket cell itself (*Figure 4J*) allows particularly evident distinction of zones in the region relevant to basket cells, we used it to test whether HCN1 basket cell zones extend beyond the limits of lobule VIII (*Figure 4*). After co-staining with HCN1 and NFH, we found that zones with high NFH expression correspond to distinct HCN1 zones in lobule VII (*Figure 4A–C*) and maintain that relationship through lobules VIII (*Figure 4D–F*) and IX (*Figure 4G–I*). We also observed that the size difference was less apparent in lobule IX compared to that in VII or VIII. This is intriguing as it suggests the possibility of additional levels of intricacy in the patterning of basket cell zones in relation to the underlying molecular, developmental, circuit, and functional complexity of the cerebellum (*Sillitoe and Joyner, 2007*). We quantified both the size of the HCN1-expressing pinceau region as well as NFH expression, which is localized to both the pinceau as well as the Purkinje cell (*Demilly et al., 2011*), in the pinceau region (*Figure 4K*). We found that the pinceau region revealed by both of these markers was larger in NFH-positive zones compared to the negative zones. Additionally, there was no overlap of pinceau size between positive and negative zones among any of the lobules included in the analysis.

## Different commonly used basket cell markers are in fact expressed in zones

In addition to HCN1, cerebellar basket cell pinceaux express a variety of molecular markers, and among these are Kv1.1 (*Wang et al., 1994*; *Iwakura et al., 2012*), PSD95 (*Fukaya and Watanabe, 2000*; *Sivilia et al., 2016*), and GAD67 (*Iwakura et al., 2012*; *Sivilia et al., 2016*). We first set out to confirm that each protein shared a similar sub-cellular compartment within the basket cells specifically in lobule VIII by co-staining with HCN1. We found that in all cases, the pinceaux were robustly co-stained and shared an identical expression localization (Kv1.1 *Figure 5B* left; PSD95 *Figure 5B* center; GAD67 *Figure 5B* right). We next tested whether these three additional markers are also heterogeneously distributed around Purkinje cells. Similar to HCN1, we found that Kv1.1, PSD95, and GAD67 all adhere to the zonal boundaries, as assessed on coronal-cut tissue sections from lobule VIII (*Figure 5A–B*). Purkinje cells with large versus small pinceaux, as defined by marker expression in the pinceau, established clear-cut boundaries (dotted lines in *Figure 5*). Interestingly, all four markers revealed an identical staining pattern; that is, all four basket cell markers delineated the same spatial expression pattern, in the same zones.

## The zonal patterning of basket cell projections is based on the size of their pinceaux

Despite their diverse functions, all four basket cell marker proteins have the same zonal pattern. This is peculiar given that the Purkinje cell map, which consists of complex arrays of interdigitating patterns, is thought to instruct the formation of its afferent microcircuits (*Miterko et al., 2018*). Therefore, unlike zebrinII, and the two dozen-plus known markers that form a molecular map, we tested the alternate possibility that perhaps basket cell zones represent a more fundamental feature of the circuit: its anatomy. To test this hypothesis, we used a genetic fate mapping approach to selectively mark basket cells and specifically highlight the boundaries of their cell membranes with a conditional reporter (*Figure 6*). We recently showed that an *Ascl1*^CreERT2^ allele can be used to mark and track basket cells based on their birth date during late embryogenesis (*Brown et al., 2019*). *Ascl1*, also known as *Mash1*, encodes a member of the basic helix-loop-helix (BHLH) family of transcription factors. A knock-in allele of *CreER* into the *Ascl1* locus faithfully reports on the differentiation of

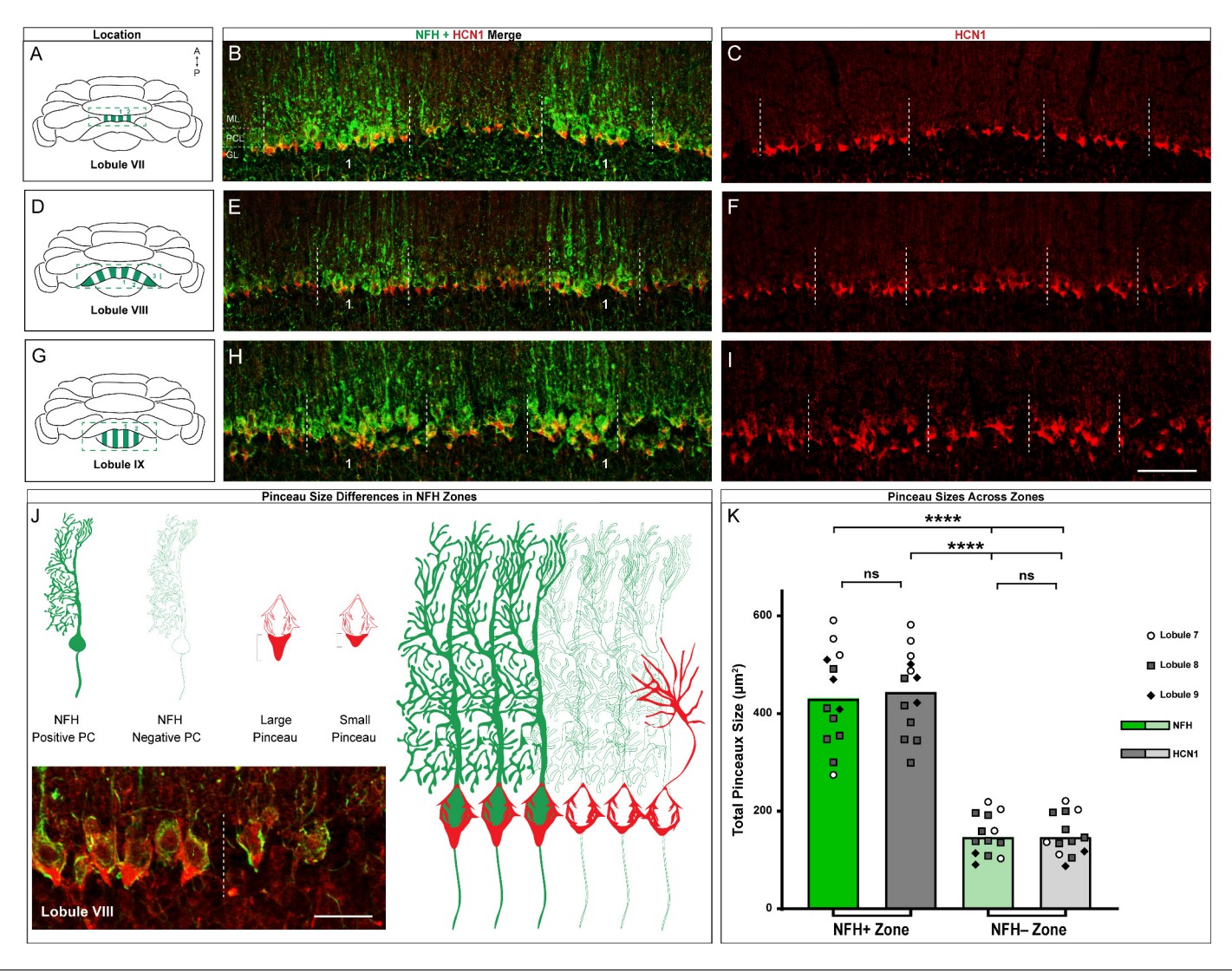

**Figure 4.** HCN1-labeled basket cell pinceaux are larger in NFH-positive Purkinje cell zones across different cerebellar lobules. (**A, D, G**) Whole-mount cerebellum schematic showing NFH expression patterns in lobules VII, VIII, and IX respectively. NFH-positive zones are marked as 1, 2, 3. (**B, C**) Coronal sections cut through lobule VII showing NFH (green, Purkinje cell) and HCN1 (red, pinceau) expression. Dotted lines delineate the Purkinje cell zonal boundaries. The Purkinje cell layer (PCL), molecular layer (ML), and granular layer (GL) are labeled as guides for locating the basket cell pinceaux. (**E, F**) Coronal sections cut through lobule VIII showing NFH and HCN1 expression. (**H, I**) Coronal section showing NFH and HCN1 expression in lobule IX. Scale bar is 100 μm. (**J**) Schematic depiction of pinceau size differences between NFH-positive (left) and -negative (right) zones, with larger pinceaux located on Purkinje cells in the NFH-positive zones. For simplicity, we did not include a schematized representation of NFH expression within the basket cell terminals (pinceaux), although its expression there should be noted (*Demilly et al., 2011*). Inset in the bottom left shows the difference between pinceau sizes in an NFH-positive (left) and -negative (right) zone, in tissue from lobule VIII stained with NFH (green) and HCN1 (red). Scale bar is 30 μm (N = 4). (**K**) Quantification of pinceau area in NFH-positive and -negative zones analyzed for HCN1 and NFH, the latter of which is expressed in the pinceaux in addition to the Purkinje cells. Pinceau in NFH-positive zones (mean = 431.2 μm$^2$ (NFH) and 444.6 μm$^2$ (HCN1), SD = 99.3 μm$^2$ (NFH) and 85.94 μm$^2$ (HCN1)) are significantly larger than those in NFH-negative zones (mean = 147.8 μm$^2$ (NFH) and 148 μm$^2$ (HCN1), SD = 42.02 μm$^2$ (NFH) and 42.87 μm$^2$ (HCN1)). Each data point indicates the total area of multiple NFH or HCN1-labeled pinceaux within a 100 μm-wide region of an NFH-positive or -negative Purkinje cell zone, reported in μm$^2$ (N = 6, n = 6 sections, 13 NFH-positive Purkinje cell zones and 13 NFH-negative Purkinje cell zones; measurements from lobule 7 (white circles), lobule 8 (gray squares), and lobule 9 (black triangles) are represented; ****p<0.0001).

The online version of this article includes the following source data for figure 4:

**Source data 1.** Source data for representative graphs in *Figure 4*.

GABAergic neurons in the cerebellum, and it has a dual function in labeling different subsets of inhibitory neurons at the time of their birth (*Sudarov et al., 2011*). Here, we crossed the *Ascl1$^{CreERT2}$*

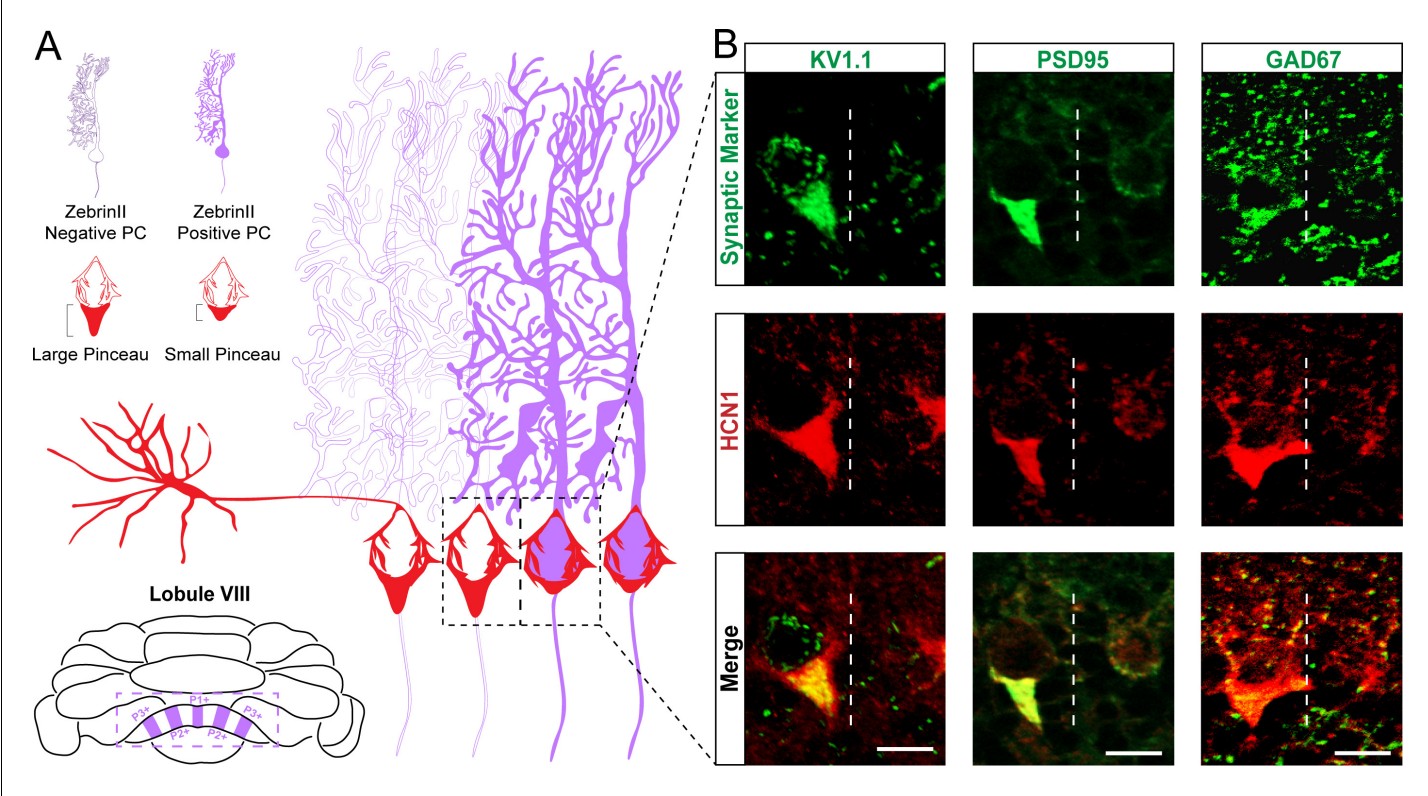

**Figure 5.** Kv1.1, PSD95, and GAD67 label basket cell pinceaux and adhere to the same zonal patterning as HCN1. (**A**) Schematic of basket cell pinceau size differences across zonal boundaries. Dotted area around Purkinje cell somata depicts the boundary between a zebrinII-negative zone (left) and a zebrinII-positive zone (right). Basket cell pinceaux are larger, on average, in the zebrinII-negative zones. (**B**) Magnified images of Kv1.1, PSD95, GAD67 (green), and HCN1 (red) expression in basket cell pinceaux across a zebrinII zonal boundary. Dotted white lines indicate the boundary between a zebrinII-negative zone (left) and a zebrinII-positive zone (right). Pinceau sizes are distinctly larger in the zebrinII-negative zone as marked by all four pinceau markers. Merged HCN1 and Kv1.1, PSD95, GAD67 expression is shown in the bottom row, respectively (*N* = 7 for Kv1.1, seven for PSD95, and seven for GAD67). Scale bars are 15 µm.

mice to a mouse line that expresses myristoylated GFP (mGFP) in differentiated neurons (*Hippenmeyer et al., 2005*), but only after recombination is induced upon tamoxifen administration to the mice (*Brown et al., 2019*). We chose this genetic marking strategy because oral gavage of tamoxifen to pregnant dams when their embryos are embryonic day (E) 18.5 labels a rich population basket cells with recombination at ~46% across the entire cerebellum (*Brown et al., 2019*; the genetic strategy is schematized in *Figure 6E*), and the mGFP reporter impressively fills the entire axons of even the finest projections in the cerebellum (*Sillitoe et al., 2009*). After inducing basket cell recombination during development, we followed the marked cells into adulthood to examine their architecture using triple-staining with a pan-Purkinje cell marker, GFP expression, and a Purkinje cell zone marker. The IP3R1 receptor uniformly marks Purkinje cells (*Figure 6A*), whereas the genetically marked basket cell pinceaux delineate a sharp boundary within the PCL (*Figure 6B*). The dotted line in *Figure 6B* separates the pinceaux into (1) a large subset, with prominent profiles around the base of the Purkinje cells and extending deeper into the GL onto the initial segment of the Purkinje cell axons (larger open bracket, left in *Figure 6B*) and (2) a small subset, with less prominent profiles, but that nevertheless adopts the same architectural connectivity with the Purkinje cells (smaller open bracket, right in *Figure 6B*). Labeling with PLCβ4 demonstrates that the division of basket cell projections respects the boundaries of the Purkinje cell zones (*Figure 6C*). However, compared to the strict and uncompromising relationship between climbing fibers and Purkinje cells (*Gravel et al., 1987*; *Voogd and Ruigrok, 2004*; *Pijpers and Ruigrok, 2006*; *Sugihara and Quy, 2007a*; *Reeber and Sillitoe, 2011*; *Reeber et al., 2013*), the basket cell-to-Purkinje cell topography is not perfect at the zonal boundaries (*Figure 6D*). It is perhaps more reminiscent of the mossy fiber-

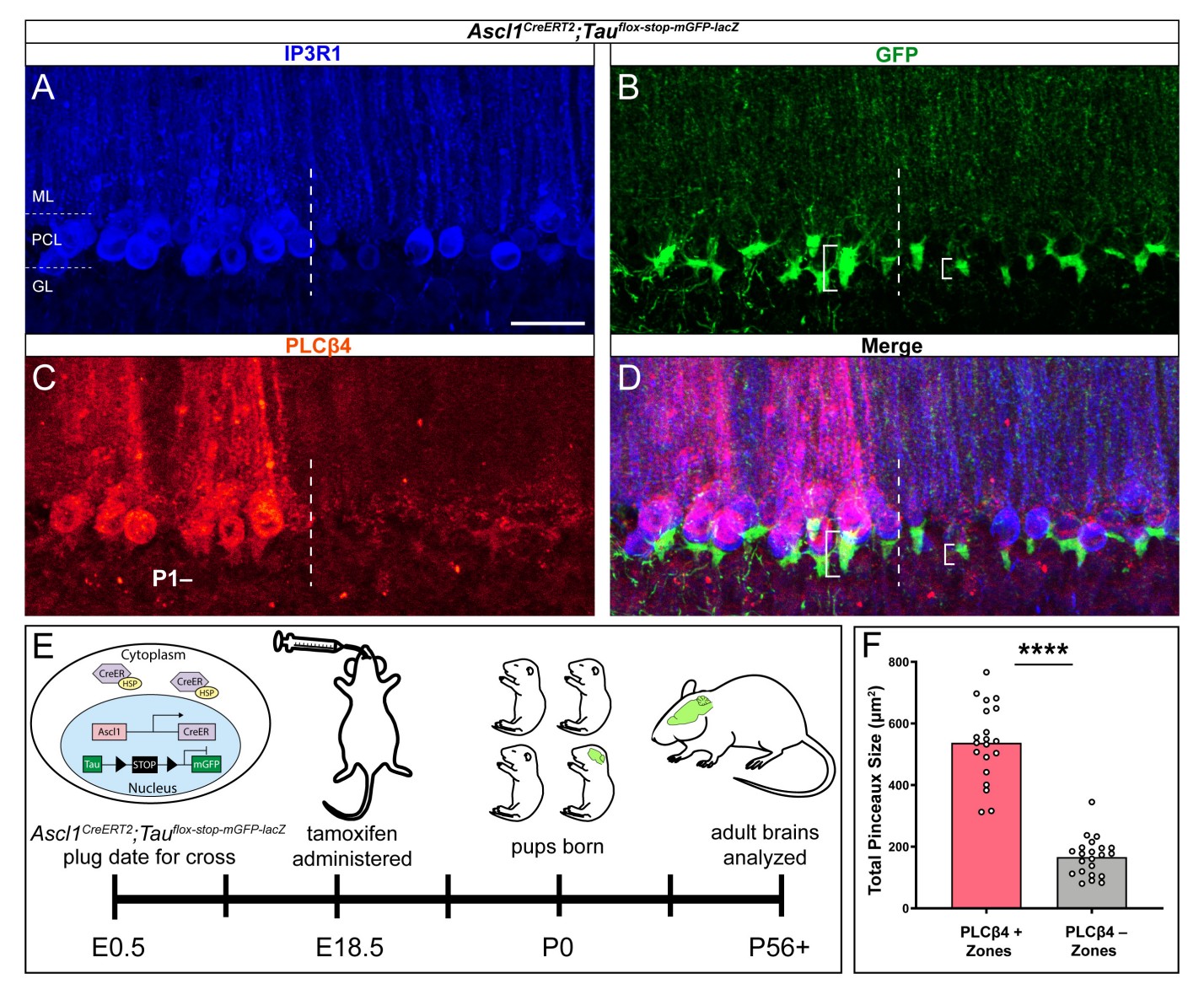

**Figure 6.** Genetically marked basket cell pinceaux are distinguished by size according to Purkinje cell zones. (A, B, C) Coronal sections cut through the cerebellar cortex showing IP3R1 (blue, PC), GFP (green, pinceau), and PLCβ4 (red, PC) expression in *Ascl1^CreERT2^;Tau^flox-stop-mGFP-lacZ^* tissue. Dotted line indicates the Purkinje cell boundary between a PLCβ4-positive (left) and PLCβ4-negative (right) zone. Scale bar in **A** is 50 μm. (D) Merged IP3R1, GFP, and PLCβ4 expression. Brackets highlight genetically labeled-pinceaux of different sizes between PLCβ4-positive and -negative zones, which is consistent with the results from the HCN1-labeled pinceaux. (E) Schematic of experimental timeline and procedure to generate genetically labeled basket cells and pinceaux. Upon tamoxifen administration, the CreER protein that was sequestered in the cytoplasm by HSP is now able to enter the nucleus and induce recombination at *loxP* sites. Neurons are marked with GFP after recombination. (F) Quantification of pinceau area across PLCβ4 zones reveals significantly higher total pinceau size in PLCβ4-positive zones (mean = 537.7 μm$^2$, SD = 125.2 μm$^2$) compared to PLCβ4-negative zones (mean = 166.5 μm$^2$, SD = 62.17 μm$^2$). Each data point indicates the total area of multiple HCN1-labeled pinceaux within a 100 μm-wide region of a PLCβ4-positive or -negative Purkinje cell zone, in μm$^2$ ($N = 4$, $n = 8$ sections, 20 PLCβ4-positive zones and 22 PLCβ4-negative zones; ****$p<0.0001$). The online version of this article includes the following source data and figure supplement(s) for figure 6:

**Source data 1.** Source data for representative graphs in *Figure 6*.
**Figure supplement 1.** Controls for assessing genetically labeled basket cells.

to-Purkinje cell topography that shows an obvious pattern of zones, although the relationship at the boundaries is more complex (*Brochu et al., 1990*; *Pakan et al., 2010*; *Sillitoe et al., 2010*; *Ruigrok, 2011*; *Reeber and Sillitoe, 2011*). Mossy fiber zones often extend beyond the boundaries

defined by the Purkinje cell zones. Still, quantification of the basket cell pinceaux using GFP fluorescence genetic marking confirms that as a population, the patterning of the pinceaux into zones reflects a significant difference in their sizes between zones (*Figure 6F*). Interestingly, the genetic marking strategy labeled collateral fibers in the GL that are also restricted to Purkinje cell zones (see GL in *Figure 6B*). The collaterals are prominent below the PLCβ4-expressing zones with little to no labeling in PLCβ4-negative zones.

## Purkinje cell neurotransmission controls the segregation of basket cell projections into zones with large and small pinceaux

The establishment of Purkinje cell zones is dependent on a sequential (but overlapping) series of mechanisms involving their birth date (*Hashimoto and Mikoshiba, 2003*; *Namba et al., 2011*), molecular identity (*Croci et al., 2006*), patterning (*Baader et al., 1999*; *Sillitoe et al., 2008a*), and cell migration (*Larouche et al., 2008*). The patterning of afferents is also dependent on these Purkinje cell molecular processes (*Sillitoe et al., 2010*). At the level of specific cell-to-cell connections, distinct molecular mechanisms also control basket cell targeting. The targeting of basket axons to the AIS depends on Semaphorin 3A (Sema3A) and its receptor, neuropilin-1 (NRP1; *Telley et al., 2016*). Sema3A is secreted by Purkinje cells, which attracts the basket cell axons that express NRP1 toward the initial segment. NRP1 also mediates sub-cellular cell-to-cell recognition through a transsynaptic interaction with neurofascin 186 (NF186), a cell adhesion molecule of the L1 immunoglobulin family that is required for the formation and maintenance of the pinceau (*Ango et al., 2004*; *Zonta et al., 2011*; *Buttermore et al., 2012*). However, even though basket cells are born during embryogenesis (see *Figure 6E*), functional basket cell connections are formed postnatally (*Sotelo, 2008*), a period when neuronal activity starts to remodel the cerebellar wiring diagram for function (*Kano and Watanabe, 2019*). Indeed, the molecular genetics and morphogenetic programs act cooperatively with neurotransmission to shape afferent patterning (*Tolbert et al., 1994*), and Purkinje cells specifically guide them into precise zones (*White et al., 2014*). We therefore tested whether Purkinje cell neurotransmission also instructs the zonal patterning of basket cell pinceaux. GABAergic neurotransmission is selectively silenced in Purkinje cells of $Pcp2^{Cre};Slc32a1^{flox/flox}$ mice (*White et al., 2014*). This particular $Pcp2^{Cre}$ allele is ideal for our purpose because it expresses *Cre* during embryogenesis and continues into adulthood (*Lewis et al., 2004*), which means that even the developing Purkinje cells lack *Slc32a1* after recombination occurs with the *floxed* allele (*Tong et al., 2008*). In these mutants, Purkinje cells are capable of receiving signals and firing simple spikes and complex spikes, although they cannot communicate their computations downstream via fast neurotransmission using GABA (*White et al., 2014*; *Stay et al., 2019*). Compared to control $Slc32a1^{flox/flox}$ mice (*Cre*-negative, no *Slc32a1* deletion; *Figure 7A,C,E*), the mutants that lack *Slc32a1* in Purkinje cells do not have a clear distinction of Purkinje cell zones or HCN1 zones, as defined by the basket cell pinceaux (*Figure 7B,D,F*). Instead, we observed a uniform distribution of HCN1, suggesting that basket cell pinceaux are all approximately the same size in the mutants. Quantification of pinceau size based on HCN1 expression confirmed that Purkinje cell neurotransmission is required for basket cell size diversity, and is the basis of their zonal plan (*Figure 7G–H*). Without Purkinje cell neurotransmission, all basket cell pinceaux were not significantly different in size compared to control pinceaux within zebrinII-positive zones (*Figure 7G–H*). Despite the blurring of Purkinje cell zonal boundaries in the $Pcp2^{Cre};Slc32a1^{flox/flox}$ mutant mice (*White et al., 2014*), we were still able to analyze zonal properties by accounting for the presence of the blurred regions. We set the zonal boundary at the very last Purkinje cell that clearly expressed the zonal marker, which effectively created a 'blurred' zonal region on one side of the defined boundary and a 'pure' zonal region on the other, which were then used for quantification. Therefore, our data showing reduced pinceau size in the zebrinII-negative zones does not represent a mixing of cellular identities in a particular region that contains the large pinceaux on zebrinII-negative Purkinje cells diluted by the small pinceaux on zebrinII-positive cells, but rather the data suggest that the zebrinII-negative Purkinje cells in the mutant are innervated by pinceaux with reduced sizes. Based on these data, we argue that Purkinje cell inhibitory neurotransmission influences basket cell diversity by sculpting pinceau structure and designating them into large versus small subsets.

We next asked whether the neurotransmission at basket cell-to-Purkinje cell synapses might also play a role in instructing the Purkinje cell zonal patterns. Specifically, we asked: if the Purkinje cell map controls both the genetic programs as well as the activity required for zonal patterning, then

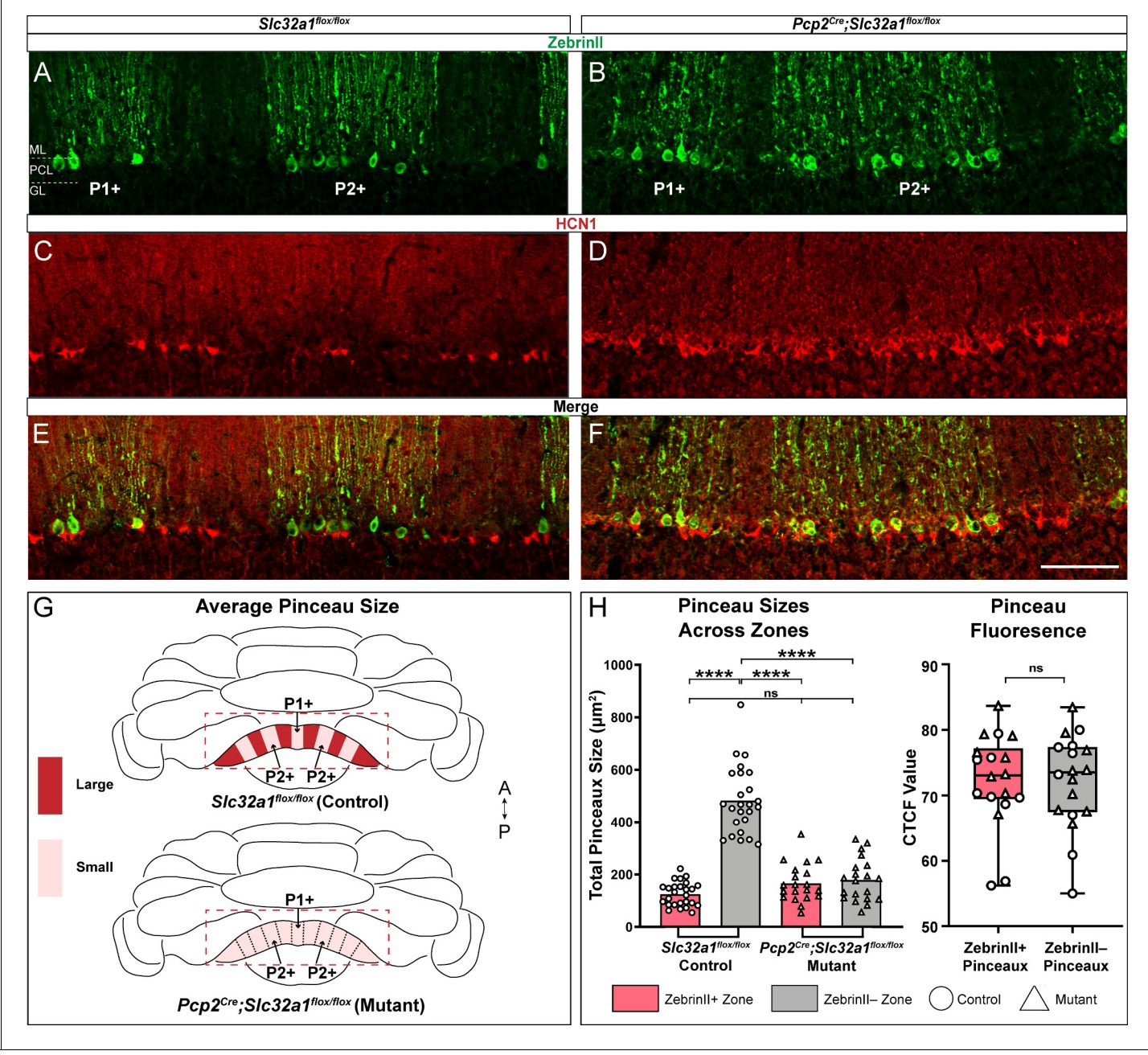

**Figure 7.** Zonal patterning of basket cell pinceaux is disrupted in *Pcp2^Cre^;Slc32a1^flox/flox^* mutants. (**A, B, C, D**) Anatomically matched coronal sections through lobule VIII showing zebrinII (green, PC) and HCN1 (red, pinceau) expression. (**A, C**) *Slc32a1^flox/flox^* (control) data. (**B, D**) *Pcp2^Cre^;Slc32a1^flox/flox^* (mutant) data, showing altered zonal organization of both Purkinje cells and pinceaux compared to controls. (**E, F**) merged zebrinII and HCN1 expression in controls and mutants, respectively. Scale bar is 100 μm. (**G**) Schematic whole-mount cerebellum diagram showing differences in pinceau size organization in *Slc32a1^flox/flox^* controls and *Pcp2^Cre^;Slc32a1^flox/flox^* mutants. In controls, pinceau sizes fall into distinct zonal domains, with dark red depicting areas with larger pinceaux and light red depicting those with smaller pinceaux. In the mutants, the zonal size organization is largely eliminated across all regions. (**H**) Left: Quantification of pinceau area across Purkinje cell zones reveals significantly smaller total pinceau size in zebrinII-positive zones (mean = 124.8 μm², SD = 45.65 μm²) compared to zebrinII-negative zones (mean = 482.6 μm², SD = 128.8 μm²; p<0.0001) in *Slc32a1^flox/flox^* controls, but there was no significant difference in pinceau sizes in *Pcp2^Cre^;Slc32a1^flox/flox^* mutants (mean = 166.5 μm², SD = 71.94 μm² for zebrinII-positive zones; mean = 178.7 μm², SD = 81.72 μm² for zebrinII-negative zones; p=0.9719). Additionally, while both mutant zones had significantly smaller pinceaux compared to control zebrinII-negative zones (control zebrinII-negative vs. mutant zebrinII-positive p<0.0001; control zebrinII-negative vs. mutant zebrinII-negative p<0.0001) there was no significant difference in the size of pinceaux between that of the mutant zones and the size of pinceaux in the control zebrinII-positive zones (control zebrinII-positive vs. mutant zebrinII-positive p=0.3883; control zebrinII-positive vs. mutant zebrinII-negative p=0.1755). Each data point indicates the total area of the ROI covered by HCN1-labeled pinceaux within a 100 μm-wide region of a

*Figure 7 continued on next page*

*Figure 7 continued*

zebrinII-positive or -negative Purkinje cell zone, in μm². For mutant mice, *N* = 4, *n* = 8 sections, 20 zebrinII-positive Purkinje cell zones and 20 zebrinII-negative Purkinje cell zones. For controls, *N* = 6 mice, 12 sections, 26 zebrinII-positive zones and 25 zebrinII-negative zones. Right: Corrected total cell fluorescence (CTCF) analysis reveals no significant difference in HCN1-labeled pinceau fluorescence intensity between pinceaux associated with zebrinII-positive (mean = 72.24, SD = 7.22) and zebrinII-negative (mean = 72.15, SD = 7.2) Purkinje cells, from both control and mutant animals. Each data point represents the CTCF value of a 1 μm² region in a single pinceau (*N* = 6 control and four mutant mice, *n* = 18 large and 18 small pinceaux; p>0.05).

The online version of this article includes the following source data for figure 7:

**Source data 1.** Source data for representative graphs in *Figure 7*.

can the afferents also contribute to the shaping of the Purkinje cell map that they integrate into? To address this question, we again used the *Ascl1^CreER* allele (*Sudarov et al., 2011*), but this time we crossed it to the *Slc32a1^flox/flox* line (*Tong et al., 2008*) in order to block inhibitory neurotransmission from basket cells to Purkinje cells by delivering tamoxifen to E18.5 pups in utero by removing the vesicular GABA transporter (VGAT) in the interneurons (*Brown et al., 2019*). We then stained Purkinje cells for zebrinII and revealed that the zonal plan (*Figure 8A*) was indistinguishable when compared to the patterns of zones in different lobules from the anterior, central, posterior, and nodular domains (*Figure 8B*) between controls (*Figure 8C*; *CreER* is not expressed because the mice do not have the allele, and as a result *Slc32a1* is left intact, although like the mutants, the control mice are also given tamoxifen) and mutants (*Figure 8D*). We next stained for both zebrinII and HCN1 to determine whether the constitutive lack of inhibitory neurotransmission from basket cells to Purkinje cells ultimately affected the zonal patterning of the pinceaux (*Figure 8E–J*). We found that patterning of basket cell pinceaux was unaffected by this lack of basket cell neurotransmission (*Figure 8K*). Therefore, inhibitory basket cell output does not control the anterior-posterior or medial-lateral patterning of molecular markers in the Purkinje cells. Nor does inhibitory basket cell neurotransmission control the general features of zonal patterning of the basket cell pinceaux. These data also confirm that Purkinje cell neurotransmission contains the necessary instructions to restrict basket cells into a highly patterned zonal map, with a key anatomical substrate of connectivity established by segregating pinceaux into distinct sizes.

## Discussion

The cerebellum is organized into a fundamental map of zones defined by molecular expression patterns, neuronal firing properties, behavioral outputs, and even disease phenotypes. Purkinje cells are at the center of each zone, receiving precisely mapped inputs from excitatory climbing fibers and mossy fibers. Here, we demonstrate that the inhibitory projections from basket cells onto Purkinje cells are also patterned into zones. We identify that HCN1, Kv1.1, PSD95, and GAD67 are all expressed in basket cell pinceaux and uncover a pattern of zones in the adult cerebellum. However, their expression reveals a unique feature of cerebellar topography: their zonal patterning is defined by the sizes of the basket cell pinceaux, rather than by spatial differences in protein expression. We tested whether Purkinje cells drive the topography of inhibitory projections, as they do for excitatory afferents. Interestingly, manipulating Purkinje cell neurotransmission eliminated the division of basket cells into large and small zones. While we found that Purkinje cell neurotransmission influences basket cell zonal patterning, we also found that neurotransmission from basket cells was not capable of similarly affecting either Purkinje or basket cell patterning. Our data demonstrate that basket cell projections are topographically organized, and that their patterning is dependent on proper neurotransmission in the cerebellar cortex. The results provide a neural substrate for how cerebellar circuitry might control module-specific firing properties and encode diverse behavioral outputs. The finer details of cerebellar patterning have been unveiled using protein expression (*Hawkes and Leclerc, 1987*), mRNA expression (*Millen et al., 1995*), viral marking (*Hashimoto and Mikoshiba, 2003*), transgenic alleles (*Furutama et al., 2010*; *Fujita et al., 2010*), and conditional genetic labeling (*Sillitoe et al., 2009*) and, in addition, the topography of afferents has been studied using injection approaches of neural tracing (*Sugihara and Quy, 2007a*), genetically encoded neural tracers (*Braz et al., 2002*), and genetically encoded reporters (*Hantman and Jessell, 2010*). However, the initial motivations to study cerebellar patterns were based purely on anatomical analyses; Jan Voogd

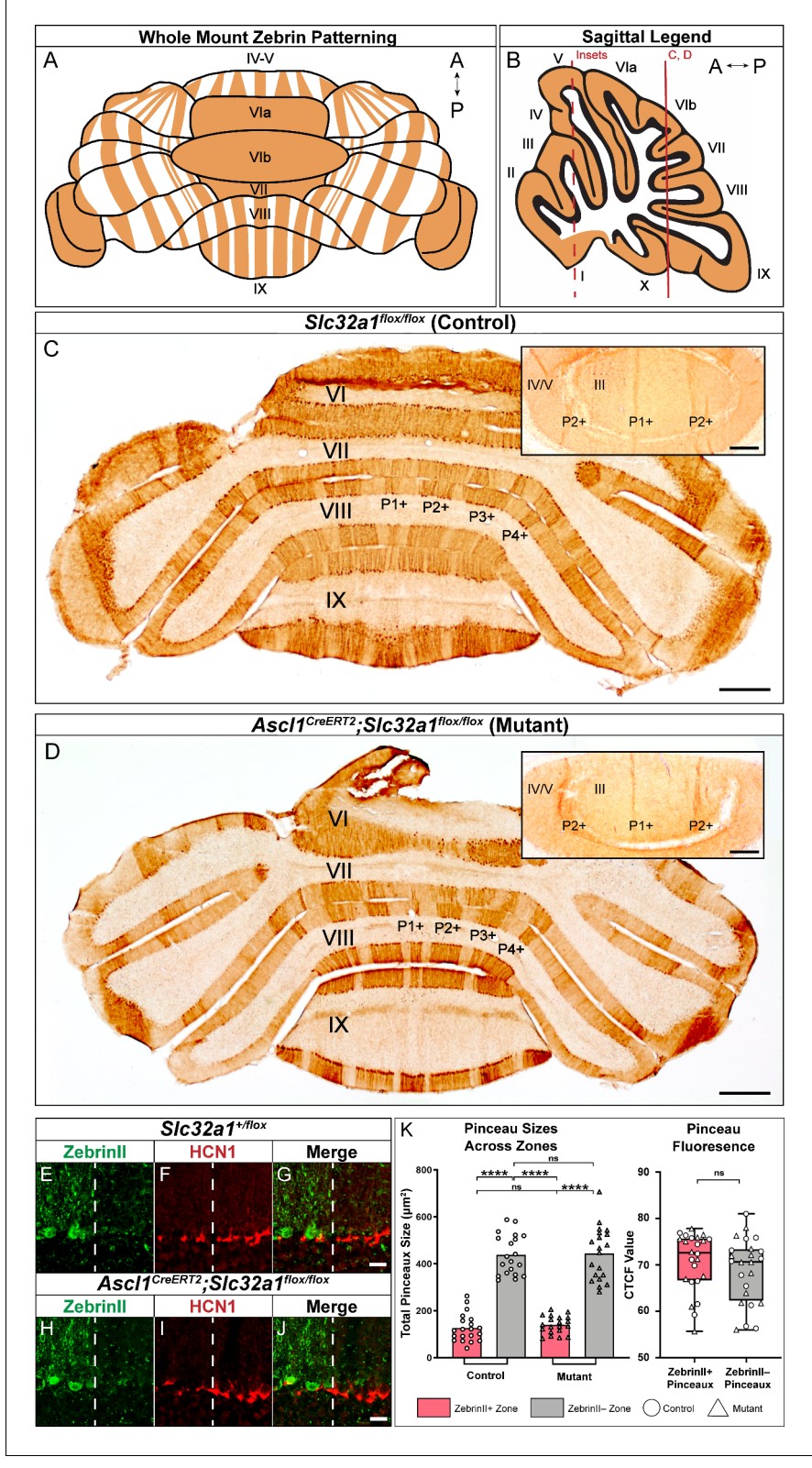

**Figure 8.** Silencing basket cell GABAergic inhibitory neurotransmission does not affect the zonal patterning of Purkinje cells. (**A**) Schematic representation of normal zebrinII patterning across the whole mouse cerebellum, seen in a whole-mount configuration. (**B**) Sagittal schematic of a mouse cerebellum slice at the midline; the red vertical line indicates the anatomical location of the coronal sections shown in **C** and **D**. The red dotted line indicates the

*Figure 8 continued on next page*

*Figure 8 continued*

location of the cerebellum that the insets were acquired from. (C) Coronal section from a control mouse given tamoxifen at E18.5, stained to reveal normal zebrinII expression patterning (*N* = 4, scale bar is 500 µm). Because the *Ascl1$^{CreERT2}$* allele was not expressed in this animal, inhibitory neurotransmission of basket cells was not affected. Inset in the top right corner shows a higher power magnification image from lobules III and IV/V in the anterior cerebellum (scale bar is 250 µm), with normal zebrinII zonal patterning for that region of the cerebellum. Coronal-cut tissue section from a mouse expressing both the *Ascl1$^{CreERT2}$* and *Slc32a1$^{flox/flox}$* alleles, given tamoxifen at E18.5 to target the silencing of neurotransmission in basket cells. Because both the *Ascl1$^{CreERT2}$* allele and the *Slc32a1$^{flox/flox}$* allele, which is used to delete *Slc32a1* with spatial and temporal control, were expressed in this animal, cerebellar basket cell neurotransmission was silenced throughout its lifetime. Despite this, staining in the anterior (top right inset), central and posterior lobules reveals that zebrinII patterning is unchanged in the absence of basket cell neurotransmission, as shown in **D** (*N* = 4). In lobules III and IV/V of the anterior cerebellum (inset), the ~500 µm distance between the P1+ and P2+ zebrinII zones (***Sillitoe and Hawkes, 2002***; ***Sillitoe et al., 2008b***) and the sharpness of the zebrinII Purkinje cell zonal boundaries is maintained after GABAergic neurotransmission is genetically blocked at the basket cell terminals (scale bar is 250 µm). (E-G) Example immunohistochemistry for quantification of pinceaux size in a *Slc32a1$^{+/flox}$* mouse. Scale = 25 µm. ZebrinII boundary = dotted line. (E) ZebrinII expression in Purkinje cells. (F) HCN1 expression in basket cell pinceaux. (G) ZebrinII and HCN1 merged image. (H-J) Example immunohistochemistry for quantification of pinceau size in an *Ascl1$^{CreERT2}$;Slc32a1$^{flox/flox}$* mouse. Scale = 25 µm. ZebrinII boundary = dotted line. (H) ZebrinII expression in Purkinje cells. (I) HCN1 expression in basket cell pinceaux. (J) ZebrinII and HCN1 merged image. (K) Left: Quantification of pinceau area across Purkinje cell zones reveals significantly smaller total pinceau size in zebrinII-positive zones compared to zebrinII-negative zones in both *Slc32a1$^{flox/flox}$* controls (mean = 127.4 µm$^2$, SD = 59.25 µm$^2$ for zebrinII-positive zones; mean = 439.1 µm$^2$, SD = 85.34 µm$^2$ for zebrinII-negative zones; p<0.0001) and *Ascl1$^{CreERT2}$;Slc32a1$^{flox/flox}$* mutants (mean = 140.6 µm$^2$, SD = 37.73 µm$^2$ for zebrinII-positive zones; mean = 443.8 µm$^2$, SD = 113.3 µm$^2$ for zebrinII-negative zones; p<0.0001). Unlike the effects seen with silenced Purkinje cells as shown in ***Figure 7***, silencing basket cells does not eliminate the size difference between pinceaux in zebrinII-positive and zebrinII-negative regions in the mutants. Each data point indicates the total area of the ROI covered by HCN1-labeled pinceaux within a 100 µm-wide region of a zebrinII-positive or -negative Purkinje cell zone, in µm$^2$. For control mice, *N* = 3, *n* = 4 sections, 20 zebrinII-positive Purkinje cell zones and 20 zebrinII-negative Purkinje cell zones. For mutants, *N* = 3 mice, *n* = 4 sections, 20 zebrinII-positive zones and 20 zebrinII-negative zones. Right: Corrected total cell fluorescence (CTCF) analysis reveals no significant difference in HCN1-labeled pinceau fluorescence intensity between pinceaux associated with zebrinII-positive (mean = 70.76, SD = 6.292) and zebrinII-negative (mean = 68.91, SD = 7.0) Purkinje cells, from both control and mutant animals. Each data point represents the CTCF value of a 1 µm$^2$ region in a single pinceau (*N* = 3 control and three mutant mice, *n* = 12 large and 12 small pinceaux per genotype; p>0.05).

The online version of this article includes the following source data for figure 8:

**Source data 1.** Source data for representative graphs in ***Figure 8***.

---

expanded on the initial finding of ***Verhaart, 1956*** who used the Häggqvist myelin stain to reveal small, medium, and large caliber axons in the brachium conjunctivum. Voogd demonstrated the presence of white matter compartments that contained large myelinated axons which were separated by narrow bands of small fibers. Some key features he studied further were the continuity of compartments across subsets of lobules, and that the compartments housed the axons of Purkinje cells that were topographically linked to specific cerebellar nuclei (***Voogd, 1964***). Within the cerebellar cortex, Hawkes and colleagues also revealed a compartmental division of the cerebellum that was based on anatomy, showing that after a particular preparation of the tissue, the GL forms 'blebs' that respect the boundaries of zebrinII expression (***Hawkes, 1997***). Our data integrates the molecular properties of basket cell pinceaux with their connectivity to Purkinje cell axons, unmasking a fundamental level of zonal patterning that segments basket cell projections based on their sizes. Interestingly, although the authors did not discuss it, PLCβ1 expression shows predominant basket cell staining particularly around PLCβ4-expressing Purkinje cells (***Fukaya et al., 2008***). We predict that markers that have a seemingly uniform expression in basket cells should in fact reveal cerebellar zones based on pinceau size, although we do not exclude the possibility that some molecules may be expressed in patterns and reveal a finer level of basket cell organization, irrespective of their zonal sizes.

The zonal topography of the pinceaux raises a critical functional question: how does basket cell heterogeneity impact cerebellar function? Multiple lines of experimental evidence using different

model systems suggest a role for zones during behavior (*Schonewille et al., 2006*; *Horn et al., 2010*; *Cerminara and Apps, 2011*; *Graham and Wylie, 2012*; *Long et al., 2018*), and these studies were supported by electrophysiological analyses indicating that synaptic plasticity may be determined by zone-specific properties (*Wadiche and Jahr, 2005*; *Paukert et al., 2010*). More recently, it has been uncovered that systematic differences in the function of zones could be hard-wired into the basic firing properties of Purkinje cells. ZebrinII-positive Purkinje cells were reported to have lower firing frequencies and fire more regularly, whereas zebrinII-negative Purkinje cells have a higher firing frequency and a more irregular pattern of activity (*Zhou et al., 2014*; *Xiao et al., 2014*). Moreover, consistent with the highly organized convergence of mossy fibers and climbing fibers within dedicated zones (*Voogd et al., 2003*), in vivo electrophysiology recordings demonstrate zone-specific interactions in simple spike and complex spike activity (*Tang et al., 2017*). Interestingly, during development there is a converse relationship such that Purkinje cell neurotransmission itself is required for precisely shaping the zones into fine-grained compartments (*White et al., 2014*). With the various classes of interneurons following the zonal scheme (*Consalez and Hawkes, 2013*), and the data presented in this study, it could be that Purkinje cells use developmental mechanisms to establish their own behaviorally relevant specializations, and for basket cells, this means their segregation into size-specific zones. It is suggested that Purkinje cell-zones may have discrete requirements during LTD (long-term depression) versus LTP (long-term potentiation) (*Wu et al., 2019*). ZebrinII-positive zones are predicted to have a major role in behaviors such as the vestibulo-ocular reflex, which is heavily dependent upon LTP, whereas behaviors such as eye-blink conditioning may be more dependent on LTD. We know that at least some portion of the eye-blink conditioning circuit is restricted to zebrinII-negative zones (*Attwell et al., 1999*; *Attwell et al., 2001*; *Mostofi et al., 2010*). By extrapolation, the large pinceaux in the zebrinII-negative zones could then serve to more strongly modulate the high frequency firing and more irregular activity of Purkinje cells during learning. Interestingly, optogenetic stimulation of basket cells in the deep paravermis of mouse lobule V/VI, a predominantly zebrinII-negative domain, strongly modulated the timing of the blink (*Heiney et al., 2014*). In addition, though, selective elimination of basket cell output results in an increase in Purkinje cell simple spike frequency (*Brown et al., 2019*). Together, these data indicate that basket cells may not necessarily set the normal firing rate of Purkinje cells, but instead might provide a custom brake. Therefore, Purkinje cells may determine the strength of their own innervation, which could ensure that the circuit is equipped to accommodate certain behaviors. Loss of *Slc32a1* in Purkinje cells obscures the zonal pattern, and therefore alters learning on rotarod assays (*White et al., 2014*). We propose that the establishment of neurotransmission and the formation of topographic patterns is tightly linked to the control of behavior in mature animals. However, we note that HCN1, Kv1.1, and PSD95 are all activity dependent (*Arimitsu et al., 2009*; *Grosse et al., 2000*; *Lu et al., 2004*; *Subramanian et al., 2019*). What, then, does the silencing of Purkinje cell inhibitory neurotransmission tell us about how basket cells acquire a non-uniform pattern/size (*Figure 7G*)? Silencing Purkinje cell GABAergic output likely abolishes the patterned distribution of basket cell projections as a consequence of masking Purkinje cell identities, resulting in the absence of pinceau specificity and an accompanying adjustment in protein expression patterns (*Figure 7A–F*).

There are several possibilities for how the adjustments in basket cell projection size might take effect when Purkinje cell neurotransmission is blocked. It could be that silencing Purkinje cells changes the convergence of basket cell axons. In control mice, 3–7 basket cells typically converge onto each Purkinje cell (*Palay and Chan-Palay, 1974*). Silencing Purkinje cell output, a physiological cue that segregates the projections into zones (*Figure 7*), could result in fewer average basket cell projections per Purkinje cell. Alternatively, the loss of Purkinje cell signals may eliminate a growth signal that either increases the extent of innervation from some fibers and/or restricts the size of others into large versus small projection domains. Moreover, it could be that the loss of Purkinje cell output does not change the average size or number of primary ascending and descending basket cell fibers; instead, the collateralization of smaller endings at the Purkinje cell initial segment may be defective (*Sotelo, 2008*), and perhaps more so in what would develop into the larger pinceaux. The mutant mice may have a lack of axonal refinement. Purkinje cell neurotransmission therefore instructs the local precision of extracerebellar and intracerebellar afferent projections (*White et al., 2014*). In vivo, it is likely that multiple steps are required for proper basket cell targeting onto Purkinje cells. The directional growth of basket cell projections from the soma to the AIS requires an ankyrinG-

dependent sub-cellular gradient of NF186 (*Ango et al., 2004*). NF186 is expressed on Purkinje cells and trans-synaptically interacts with neuropilin-1 (NRP1), a Semaphorin receptor expressed by basket cells, to control the formation of pinceau synapses (*Telley et al., 2016*). Here, we show that there is an added level of specificity, in a process that restricts pinceau formation according to size. We argue that Purkinje cell neurotransmission controls the distinction of basket cells by size, and although basket cell GABAergic function contributes to postnatal climbing fiber synapse elimination (*Nakayama et al., 2012*), basket cell neurotransmission does not play a role in patterning Purkinje cell zones (*Figure 8*).

The differences in baseline Purkinje cell firing rate between zebrinII-positive and zebrinII-negative zones (*Zhou et al., 2014*; *Xiao et al., 2014*) raised the intriguing possibility that the establishment of zonal patterns as defined by pinceaux size may be driven by the level of Purkinje cell neurotransmission. We expanded on this idea using a constitutive deletion of *Slc32a1* that removes fast GABAergic neurotransmission from both zebrinII-positive and zebrinII-negative Purkinje cells, a manipulation that does not eliminate the different cerebellar cell types (*White et al., 2014*). Removing Purkinje cell neurotransmission resulted in a lack of pinceau zonal organization wherein, instead of large pinceaux innervating the zebrinII-negative zones and small pinceaux localizing to the zebrinII-positive zones, there were only small pinceaux throughout both classes of zones. This change in innervation is potentially mediated by a developmental patterning event. Purkinje cell neurotransmission is normally present at the time of basket cell progenitor migration and synaptogenesis (*Leto et al., 2016*). Interestingly, the migration of the progenitors that give rise to ML interneurons, such as the basket cells, are responsive to neurotransmission, with a portion of these signals potentially arising from Purkinje cells (*Wefers et al., 2017*). Additionally, on the one hand basket cell pinceau formation and targeting is thought to depend on a gradient of NF186 expression, which in turn is dependent on the expression of AIS protein ankyrinG (*Ango et al., 2004*). On the other hand, AIS location is plastic, such that greater levels of excitation can lead to a distal shift of the AIS away from the soma (*Grubb and Burrone, 2010*). Therefore, it is possible that pinceau zonal organization is established during the differentiation of Purkinje cell electrophysiological properties, which culminates in a greater rate of neurotransmission in zebrinII-negative zones and a lesser rate in zebrinII-positive zones, with this compartmentalized relationship maintained into adulthood. Ultimately, the establishment of pinceau zones by Purkinje cells reflects an intimate inter-cellular relationship involving precise anatomical connectivity and circuit function.

Our data indicate that the maturation and maintenance of zonal Purkinje cell neurotransmission are critical for establishing basket cell pinceau size, findings that lead to the interesting hypothesis that this relationship may also influence pathology in degenerative diseases that affect the cerebellum. In control mice, the rate and pattern of Purkinje cell neurotransmission reaches maturity around the fourth postnatal week (*Arancillo et al., 2015*). However, numerous cerebellar circuit refinements occur in the postnatal weeks before this electrophysiological maturity is achieved. For example, climbing fiber synapse strengthening and parallel fiber synapse elimination continue until ~P30 (*Kano et al., 2018*), Purkinje cell dendritic remodeling continues into the third postnatal week (*Kaneko et al., 2011*) and zebrinII map refinement continues until ~P25 (*Tano et al., 1992*). Mice that exhibit Purkinje cell abnormalities and loss during this electrophysiological maturation process also exhibit abnormalities in pinceau size. For example, Purkinje cell degeneration begins at about P15 in the *Purkinje cell degeneration* (*pcd*) mutant mouse (*Landis and Mullen, 1978*; reviewed in *Wang and Morgan, 2007*) and at around P8 in the *Lurcher* mouse (*Caddy and Biscoe, 1979*). Both mutants lose more than 90% of their Purkinje cells before P30; therefore, before rates and patterns of Purkinje cell neurotransmission have fully matured. Abnormally reduced pinceau size has been noted in both of these mutant mice (*Sotelo and Alvarado-Mallart, 1987*; *Dumesnil-Bousez and Sotelo, 1993*). Additionally, pinceau size is only modestly rescued in these animals by the introduction of grafted Purkinje cells (*Sotelo and Alvarado-Mallart, 1987*; *Dumesnil-Bousez and Sotelo, 1993*). This is despite normal bioelectrical properties (*Gardette et al., 1988*) and the presence of zebrin-expressing compartments within grafted Purkinje cells (*Rouse and Sotelo, 1990*). In contrast, in cases where the majority of Purkinje cell loss occurs after the differentiation and maturation of neurotransmission or a larger proportion of Purkinje cells remain intact in the circuit, 'empty baskets' with seemingly normal pinceaux sizes remain (*Sotelo and Triller, 1979*; *Smeets et al., 2015*). This suggests a role for the establishment of mature zonal Purkinje neurotransmission during the formation of appropriately sized pinceaux. However, later insults to Purkinje cells can also affect basket

size. For example, 'prominent' baskets with 'complex' morphology have been observed in spinocerebellar ataxia type 6, SCA6 (*Lee et al., 2019*) and 'hairy baskets' have been observed in post-mortem tissue from individuals with essential tremor (*Erickson-Davis et al., 2010*). Both of these motor diseases appear late in life and have been associated with significant Purkinje cells loss. The abnormal electrophysiological properties of Purkinje cells prior to their loss in these disorders is currently a subject of intense study (*Watase et al., 2008*; *Jayabal et al., 2016*; *Kralic et al., 2005*; *Brown et al., 2020*; *Pan et al., 2020*). Whether the loss of Purkinje cell neurotransmission coupled with the loss of the cells themselves or a disruption in their normal communication properties prior to cell death affects the organization of pinceaux zones per se has not yet been tested. Regardless, there is compelling evidence that Purkinje cell neurotransmission properties have an intriguing influence over basket cell pinceau morphology and size.

Regardless of their normal functions or potential pathophysiological contributions, the electrical and chemical connectivity of the ML interneurons are highly structured, with connectivity clustering coefficients that reflect a spatial arrangement in sagittal rows (*Rieubland et al., 2014*). Electrical connections tether rodent basket cells into groups of 5 (*Alcami and Marty, 2013*). It could be that the local electrical networking, their arrangement into rows, and their size selectivity fall into a singular map, following the 'one-map hypothesis' proposed by *Apps and Hawkes, 2009*. It is interesting to speculate how such a model could benefit from basket cell patterns. Cortical output is modulated by climbing fiber and parallel fiber input as well as the intrinsic firing of Purkinje cells. However, since basket cells contribute to the excitation/inhibition (E/I) balance, and since glutamate spillover from climbing fibers impacts ML interneuron function (*Szapiro and Barbour, 2007*), it is possible that the different sizes of basket cell projections (namely their pinceaux, although likely their full innervation) uniquely complement the excitatory innervation. Together, they could drive cerebellar module function (*Wu et al., 2019*) and synchronous activity (*Welsh et al., 1995*), but also direct the precision of synaptic plasticity (*Wadiche and Jahr, 2005*; *Paukert et al., 2010*).

## Conclusions

Cerebellar basket cells are a class of ML interneurons that project to Purkinje cells. We found using several different molecular markers that basket cell pinceaux are organized into zones that coincide with the pattern of a well-established Purkinje cell map. We used an *Ascl1*$^{CreERT2}$ genetic inducible allele to leverage the spatial and temporal pattern of inhibitory interneuron development in order to mark the terminal field topography of basket cells. We reveal that basket cells are patterned according to the size of their pinceaux, which innervate the Purkinje cell AIS. Additionally, we found that Purkinje cell GABAergic neurotransmission – but not basket cell GABAergic neurotransmission – is required for the cell non-autonomous patterning of basket cell pinceaux. This study uncovers a fundamental zonal architecture of cerebellar interneuron projections and illustrates that basic neuroanatomical connectivity provides the underlying guiding principle for organizing the brain.

# Materials and methods

## Animal maintenance

Mouse husbandry and experiments were performed under an approved Institutional Animal Care and Use Committee (IACUC) protocol at Baylor College of Medicine (BCM). Male and female mouse genetic models (see below the details for the different alleles) were obtained from The Jackson Laboratory (Bar Harbor, ME, USA) and a colony was established and thereafter maintained in house at BCM. We bred mice using standard timed pregnancies, and noon on the day a vaginal plug was detected was considered embryonic day (E) 0.5. The day of birth was designated as postnatal day (P) 0. Mice of both sexes were studied. All mice used in this study were mature adults, with their ages ranging between 3 to 14 months old.

## Genetically engineered mouse lines

Three mouse lines were intercrossed to generate the alleles used in this study. The first line exhibits silenced Purkinje cell neurotransmission by elimination of the VGAT (*Slc32a1*, also known as *Vgat*, *Slc32a1*$^{tm1Lowl}$/J; The Jackson Laboratory, Bar Harbor, ME, USA, Stock No.: 012897) under the control of the Purkinje cell-specific *Pcp2* (also known as *L7*, Tg(Pcp2-cre)1Amc/J; The Jackson

Laboratory, Bar Harbor, ME, USA, Stock No.: 006207) promoter. $Pcp2^{Cre}$;$Slc32a1^{flox/flox}$ mice were generated as previously described (*White et al., 2014*). $Pcp2^{Cre}$ mice (*Lewis et al., 2004*) were crossed with a conditional 'floxed' allele of $Slc32a1$ (*Tong et al., 2008*). $Slc32a1$ is widely expressed in GABAergic and glycinergic neurons in the brain and it is essential for loading GABA into presynaptic vesicles for fast inhibitory neurotransmission (*McIntire et al., 1997*; *Chaudhry et al., 1998*; *Fujii et al., 2007*; *Saito et al., 2010*). Genotyping for the $Pcp2^{Cre}$ allele was performed using standard *Cre* primers (*Sillitoe et al., 2008a*; *Sillitoe et al., 2010*), and genotyping for the $Slc32a1^{flox}$ allele was performed according to *Tong et al., 2008*. The control mice used for the genetic manipulations were littermate controls from the $Slc32a1^{flox}$ strain, lacking *Cre* and therefore with preserved $Slc32a1$ functioning, and are referred to as $Slc32a1^{flox/flox}$ in this study. The second mouse line has a genetically encoded fluorescent tag that we used to determine the size of projections. The mice have myristoylated green fluorescent protein (mGFP) knocked-in to the Tau locus (*Hippenmeyer et al., 2005*) with an upstream floxed transcriptional stop cassette (129P2-$Mapt^{tm2Arbr}$/J; The Jackson Laboratory, Bar Harbor, ME, USA, Stock No.: 021162) as well as a knock-in allele of the $CreER^{T2}$ cassette under the control of the $Ascl1$ (also known as $Mash1$; $Ascl1^{tm1.1(Cre/ERT2)Jejo}$/J; The Jackson Laboratory, Bar Harbor, ME, USA, Stock No.: 012882) promoter ($Ascl1$-$^{CreERT2}$;$Tau^{flox-stop-mGFP-lacZ}$). To genetically label basket cells specifically, tamoxifen was administered to pregnant dams at E18.5, a time point at which subsets of basket cells emerge during embryogenesis (*Sudarov et al., 2011*). Genotyping procedures for the $Ascl1^{CreERT2}$ and the $Tau^{mGFP}$ alleles were performed according to the protocols described in *Sillitoe et al., 2009*. The third line of mice exhibits silenced basket cell inhibitory neurotransmission by elimination of $Slc32a1$ under the control of the $Ascl1$ promoter driving $CreER^{T2}$ expression ($Ascl1^{CreERT2}$;$Slc32a1^{flox/flox}$). To selectively target the deletion of $Slc32a1$ in only basket cells, tamoxifen was administered to pregnant dams at E18.5. Genotyping for the $Slc32a1^{flox}$ conditional allele was performed according to a standard polymerase chain reaction protocol as described in *Brown et al., 2019* and originally developed by *Tong et al., 2008*. Additional C57BL/6J (The Jackson Laboratory, Bar Harbor, ME, USA, Stock No.: 000664) controls were used for the initial analyses of patterns.

## Cre induction

Tamoxifen (Sigma-Aldrich catalog #T5648) was dissolved on a rocker at 37°C overnight in fresh corn oil (not older than 5 months old, stored in the dark at room temperature) at a concentration of 20 mg/ml (*Sillitoe et al., 2009*; *Zervas et al., 2004*). An 18-gauge needle was fitted onto a Luer-Lok syringe, which was used to gently pipette the solution up and down 3–5 times in order to dissolve any remaining clumps of tamoxifen. To improve pup survival when targeting the basket cells, we administered a mixture of 200 µg/g tamoxifen supplemented with 50 µg/g progesterone to the pregnant dams by oral gavage at E18.5 (*Sudarov et al., 2011*; *Bowers et al., 2012*). The full procedure for targeting the basket cells with tamoxifen was described in *Brown et al., 2019*. We tested the reliability of detecting the genetically marked cells by examining the cerebella of $CreER^{T2}$-negative mice (*Figure 6—figure supplement 1*).

## Immunohistochemistry

Perfusion and tissue fixation were performed as previously described (*Sillitoe et al., 2008a*). Briefly, mice were anesthetized by intraperitoneal injection with Avertin (2, 2, 2-Tribromoethanol, Sigma-Aldrich catalog # T4). Cardiac perfusion was performed with 0.1 M phosphate-buffered saline (PBS; pH 7.4), then by 4% paraformaldehyde (4% PFA) diluted in PBS. For cryoembedding, brains were post-fixed at 4°C for 24 to 48 hr in 4% PFA and then cryoprotected stepwise in sucrose solutions (15 and 30% diluted in PBS) and embedded in Tissue-Tek O.C.T. compound (Sakura Finetek USA; catalog #4583). Tissue sections were cut on a cryostat with a thickness of 40 µm and individual free-floating sections were collected sequentially and immediately placed into PBS. Our procedures for immunohistochemistry on free-floating frozen cut tissue sections have been described extensively in previous work (*Sillitoe et al., 2003*; *Sillitoe et al., 2010*; *White and Sillitoe, 2013*; *White et al., 2014*; *White and Sillitoe, 2017*). However, below we describe the reagents used in this study. After completing the staining steps, the tissue sections were placed on electrostatically coated glass slides and allowed to dry.

## Purkinje cell zone and basket cell projection markers

Monoclonal anti-zebrinII (*Brochu et al., 1990*) was used directly from spent hybridoma culture medium at a concentration of 1:250 (gift from Dr. Richard Hawkes, University of Calgary). ZebrinII recognizes an antigen on the aldolase C protein (*Ahn et al., 1994*) and it is a well-established marker for Purkinje cell zones. Rabbit polyclonal anti-phospholipase C β4 (PLCβ4; 1:150; Santa Cruz Biotechnology; catalog #sc-20760) was used to label Purkinje cell zones that are complementary to those revealed by zebrinII (*Sarna et al., 2006*). Neurofilament heavy chain (NFH) is also expressed in Purkinje cell zones, although it shows an additional level of zonal complexity (*Demilly et al., 2011*; *White and Sillitoe, 2013*). Mouse monoclonal anti-NFH (1:1,000; MilliporeSigma; catalog #NE1023) was used to label the soma, dendrites, and axons of adult Purkinje cells, as well as the axons and terminals of basket cells. We also used goat polyclonal anti-inositol 1,4,5-trisphosphate receptor type 1 (IP3R1; 1:250; Santa Cruz Biotechnology; catalog #sc-6093) and rabbit polyclonal anti-calbindin (1:1,000; Swant; catalog #300) as general markers to label all adult Purkinje cells. Rabbit polyclonal anti-hyperpolarization-activated cyclic nucleotide-gated channel 1 (HCN1; 1:350; Synaptic Systems), was used to label basket cell axons and pinceau terminals. Postsynaptic density protein 95 (PSD95) has been shown to have high expression in the presynaptic plexus of cerebellar basket cells (*Kistner et al., 1993*) and therefore mouse monoclonal anti-PSD 95 (1:500; UC Davis/NIH NeuroMab Facility; catalog #75–028) was used as another marker of basket cell projections. Potassium voltage-gated channel subfamily A member 1 (Kv1.1) is abundantly expressed in cerebellar basket cell axon terminals (*Laube et al., 1996*). Rabbit polyclonal anti-Kv1.1 (1:500; Alomone Labs; catalog #APC-009) was also used as marker of basket cell axons and terminals. Some tissue sections were double, triple, or quadruple-labeled with the different markers listed above, and in some cases with chicken anti-GFP (1:1,000; Abcam, catalog #AB13970) in order to visualize the mGFP reporter expression.

We visualized immunoreactive complexes either using diaminobenzidine (DAB; 0.5 mg/ml; Sigma) or fluorescent secondary antibodies. For the DAB reaction, we used horseradish peroxidase (HRP)-conjugated goat anti-rabbit and goat anti-mouse secondary antibodies (diluted 1:200 in PBS; DAKO) to bind the primary antibodies. Antibody binding was revealed by incubating the tissue in the peroxidase substrate 3,3′-diaminobenzidine tetrahydrochloride (DAB; Sigma-Aldrich, catalog #D5905), which was made by dissolving a 100 mg DAB tablet in 40 ml PBS and 10 μL 30% $H_2O_2$. The DAB reaction was stopped with PBS when the optimal color intensity was reached. Staining for fluorescent immunohistochemistry was performed using donkey anti-mouse, anti-rabbit, or anti-guinea pig secondary antibodies conjugated to Alexa-350, –488, −555, and −647 fluorophores (1:1500 for all; Invitrogen). Tissues sections were coverslipped using either Entellan mounting media (for DAB; Electron Microscopy Sciences) or FLUORO-GEL with Tris buffer (Electron Microscopy Sciences). We tested the specificity of the secondary antibodies by processing the tissue in the absence of primary antibodies. No signal was detected indicating that the staining we observed in basket or other cells was not due to non-specific signals from the Alexa or HRP-conjugated antibodies. There was also no staining when the secondary antibodies were left out of the staining mixture (*Figure 2—figure supplement 1*). Sample size was not determined using a priori power analysis, but was based on the criteria for significance in observations. A total of 56 cerebella from four genotypes of mice were used in this study, which were processed for immunohistochemistry to examine pinceau patterning (detailed numbers of animals used for specific genotypes and cellular marker combinations are listed in the figure legends). From these 56 cerebella, images from 15 controls (*Slc32a1$^{flox/flox}$* and *Slc32a1$^{+/flox}$*), four with genetically labeled basket cells (*Ascl1$^{CreERT2}$;Tau$^{flox-stop-mGFP-lacZ}$*), four with silenced Purkinje cell neurotransmission (*Pcp2$^{Cre}$;Slc32a1$^{flox/flox}$*), and three with genetically induced silencing of basket cell neurotransmission (*Ascl1$^{CreERT2}$;Slc32a1$^{flox/flox}$*) were analyzed for pinceau size and fluorescence differences using the quantification methods described below. An additional four cerebella from C57BL/6J mice were used for immunostaining controls (shown in the figure supplements). In a previous study, we showed a blurring of Purkinje cell zonal boundaries in the *Pcp2$^{Cre}$; Slc32a1$^{flox/flox}$* mice (*White et al., 2014*). Here, we analyzed cerebellar zonal properties while accounting for the presence of blurred domains by setting the zonal boundary at the very last Purkinje cell that clearly expressed the zebrinII zonal marker, which established a 'blurred' zonal region on one side of the defined boundary and a 'pure' zonal region on the other. Defining the zonal patterns in this manner aided the quantification of zonal defects in the mutant mice.

## Golgi-Cox staining

The brains from six control mice were removed from the skull and then processed using the FD Rapid Golgi Stain Kit (PK 401 from FD Neurotechnologies, INC). We focused the anatomy on optimally stained brains. All steps were carried out according to the manufacturers' instructions. After staining, the tissue was dehydrated in an ethanol series, cleared with xylene, and then mounted onto electrostatically coated glass slides with cytoseal.

## Imaging of immunostained tissue sections

Photomicrographs of stained tissue sections were captured with a Zeiss AxioCam MRm (fluorescence) and AxioCam MRc5 (DAB-reacted tissue sections) cameras mounted on a Zeiss Axio Imager. M2 microscope or on a Zeiss AXIO Zoom.V16 microscope. Apotome imaging (Apotome.2, Zeiss) of tissue sections was performed and images acquired and analyzed using either Zeiss AxioVision software (release 4.8) or Zeiss ZEN software (2012 edition). After imaging, the raw data was imported into Adobe Photoshop CC 2019 and corrected for brightness and contrast levels. The schematics were drawn in Adobe Illustrator CC 2019 and then imported into Photoshop to construct the full image panels.

## Quantification of the sizes of basket cell projections in Purkinje cell zones

Basket cell pinceau sizes within Purkinje cell zones were quantified using the Fiji distribution of ImageJ software (*Schindelin et al., 2012*). Images of mGFP-tagged basket cell projections (mainly the 'basket' portion of the projection that sits at the base of the Purkinje cell soma and the obvious pinceau terminal projection that resides on the Purkinje cell AIS) or HCN1-stained pinceaux overlaid with Purkinje cell zonal markers (ZebrinII, PLCβ4, or NFH) were loaded into Fiji. Purkinje cell zonal boundaries within the image were identified by an experimenter trained to examine cerebellar anatomy and cellular architecture. For the purposes of this study, the zonal boundaries were defined as the abutting region(s) that comes directly after the last Purkinje cell that expresses a given zonal marker. The Purkinje cell zonal marker channel was then removed from the image so that only the mGFP-tagged basket cell projections remained. Each image was subsequently set to a threshold of 19–20%, or until all baskets were clearly filled in the image. A 100 μm region of interest (ROI) containing only the mGFP-tagged baskets was selected from the previously marked Purkinje cell zone border, and the total area of the ROI covered by pinceaux was calculated using the 'analyze particles' function. This method of defining the ROI, using combined molecular expression and anatomy to identify a zonal boundary paired with analyses that consider a standardized and uniform ROI area across all images, served to limit the potential for bias in our analyses. The resulting total basket-containing area within each analyzed zone was recorded in MS Excel. Two-sample t-tests comparing total basket areas between positive and negative Purkinje cell marker zones as well as graphical representations of the statistical results were generated using GraphPad Prism software version 7 (GraphPad Software, Inc). Descriptive statistics are listed in the figure legends for the relevant figures. For the control (C57BL/6J and *Slc32a1*$^{flox/flox}$) mice, 24 coronal cerebellar sections containing a total of 52 zebrinII-positive zones and 51 zebrinII-negative zones collected from 12 different mice were analyzed. In mice with genetically labeled basket cells (*Ascl1*$^{CreERT2}$;*Tau*$^{flox-stop-mGFP-lacZ}$), eight coronal cerebellar sections containing a total of 20 PLCβ4-positive and 22 PLCβ4-negative zones from four different mice were analyzed. For the mice with silenced Purkinje cell neurotransmission (*Pcp2*$^{Cre}$;*Slc32a1*$^{flox/flox}$), eight coronal cerebellar sections containing a total of 20 zebrinII-positive zones and 20 zebrinII-negative zones from four different mice were analyzed. Values were recorded in Microsoft Excel software, and the raw data was subsequently processed through GraphPad Prism software to conduct the statistical calculations and generate the graphical representations that show the data. Unpaired two-sample, two-tailed t-tests were used when comparing two groups. Two-way ANOVAs with the Tukey-Kramer test to account for multiple comparisons were used for comparisons of more than two groups.

## Measurement and quantification of HCN1 intensity in basket cell projections

The difference in HCN1 fluorescence intensity between large and small pinceaux in control and *Pcp2^Cre^;Slc32a1^flox/flox^* mutant tissues were analyzed using ImageJ software. A total of 72 large and small basket cell pinceaux from 16 different animals (6 C57BL/6J controls, 6 *Slc32a1^flox/flox^* controls, and 4 *Pcp2^Cre^;Slc32a1^flox/flox^* mutants) were analyzed for corrected total cell fluorescence (CTCF) values. Each image was captured at 20x magnification, and analysis was focused on lobules VII-IX where the different basket cell sizes are particularly clear and easily tracked for analysis. Pinceaux in both zebrinII-positive and -negative zones were evenly selected across the different zones for fluorescence analysis. CTCF values were calculated by subtracting the product of the area of the ROI within each basket and the mean pixel value of the image background, from the summed pixel values within the ROI (Integrated Density), defined and written as:

$$\mathrm{CTCF} = (\textit{Integrated Density}) - (\mathrm{Area\ of\ ROI} \times \textit{Mean Background Fluorescence})$$

The ROI that we selected for each basket cell was kept consistent within the image at 9 pixels or 1 $\mu m^2$, as this allowed for the ROI to be small enough to fit within every basket in the image. Background fluorescence for each image was set to the pixel value of a 1 $\mu m^2$ region where there appeared to be a lack of fluorescence. Values were recorded in Microsoft Excel software, and the raw data was subsequently processed through GraphPad Prism software to conduct the statistical calculations and to generate the graphical representations that show the data. Unpaired two-sample, two-tailed t-tests were used when comparing two groups. Two-way ANOVA's with the Tukey-Kramer test to account for multiple comparisons were used for comparisons that involved more than two groups.

## Acknowledgements

This work was supported by funds from Baylor College of Medicine (BCM) and Texas Children's Hospital. RVS received support from The Bachmann-Strauss Dystonia and Parkinson Foundation, Inc, The Caroline Wiess Law Fund for Research in Molecular Medicine, The Hamill Foundation, BCM IDDRC U54HD083092, National Center for Research Resources C06RR029965, and the National Institutes of Neurological Disorders and Stroke (NINDS) R01NS089664 and R01NS100874. The BCM IDDRC Neuropathology Core performed the immunohistochemistry and histology experiments. The content is solely the responsibility of the authors and does not necessarily represent the official views of the National Center for Research Resources or the National Institutes of Health. AMB received support from F31NS101891 and MA was supported by a postdoctoral award from the National Ataxia Foundation (NAF).

## Additional information

### Competing interests

Roy V Sillitoe: Reviewing editor, *eLife*. The other authors declare that no competing interests exist.

### Funding

| Funder | Grant reference number | Author |
| --- | --- | --- |
| Baylor College of Medicine | | Joy Zhou<br>Amanda M Brown<br>Elizabeth P Lackey<br>Marife Arancillo<br>Tao Lin<br>Roy V Sillitoe |
| Texas Children's Hospital | | Joy Zhou<br>Amanda M Brown<br>Elizabeth P Lackey<br>Marife Arancillo<br>Tao Lin<br>Roy V Sillitoe |

| | | |
|---|---|---|
| Bachmann-Strauss Dystonia and Parkinson Foundation | | Roy V Sillitoe |
| National Center for Research Resources | C06RR029965 | Roy V Sillitoe |
| National Institute of Neurological Disorders and Stroke | R01NS089664 | Roy V Sillitoe |
| National Institute of Neurological Disorders and Stroke | R01NS100874 | Roy V Sillitoe |
| National Ataxia Foundation | Postdoctoral Award | Marife Arancillo |
| National Institutes of Health | F31NS101891 | Amanda M Brown |

The funders had no role in study design, data collection and interpretation, or the decision to submit the work for publication.

### Author contributions

Joy Zhou, Conceptualization, Formal analysis, Validation, Investigation, Visualization, Methodology, Writing - original draft, Writing - review and editing; Amanda M Brown, Conceptualization, Formal analysis, Validation, Investigation, Visualization, Methodology, Writing - review and editing; Elizabeth P Lackey, Validation, Investigation, Writing - review and editing; Marife Arancillo, Conceptualization, Validation, Investigation, Writing - review and editing; Tao Lin, Validation, Investigation; Roy V Sillitoe, Conceptualization, Resources, Supervision, Funding acquisition, Validation, Investigation, Writing - original draft, Project administration, Writing - review and editing

### Author ORCIDs

Joy Zhou (ID) https://orcid.org/0000-0003-1731-8800
Amanda M Brown (ID) https://orcid.org/0000-0002-1484-8972
Roy V Sillitoe (ID) https://orcid.org/0000-0002-6177-6190

### Ethics

Animal experimentation: This study was performed in strict accordance with the recommendations in the Guide for the Care and Use of Laboratory Animals of the National Institutes of Health. All animals were housed in an AALAS-certified facility on a 14hr light cycle. Husbandry, housing, euthanasia, and experimental guidelines were reviewed and approved by the Institutional Animal Care and Use Committee (IACUC) of Baylor College of Medicine (protocol number: AN-5996).

### Decision letter and Author response

Decision letter https://doi.org/10.7554/eLife.55569.sa1
Author response https://doi.org/10.7554/eLife.55569.sa2

## Additional files

### Supplementary files

• Transparent reporting form

### Data availability

All data generated or analyzed during this study are included in the manuscript and supporting files. Source data files have been provided for Figures 2, 4, 6, 7, and 8.

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
