## [Decision Letter]

**Acceptance summary:**

This study nicely describes a previously uncharacterized patterning of basket cell pinceaux size based on Purkinje cell zones. The data show that the size of basket cell pinceaux differs with zones, and suggests that this is dependent on Purkinje cell synaptic transmission.

**Decision letter after peer review:**

Thank you for submitting your manuscript "Purkinje cell neurotransmission patterns cerebellar basket cells into zonal modules defined by distinct pinceau sizes" for consideration by *eLife*. Your article has been reviewed by three peer reviewers, one of whom is a member of our Board of Reviewing Editors, and the evaluation overseen by K VijayRaghavan as the Senior Editor. The reviewers and the Reviewing Editor drafted this decision letter to help you prepare a revised submission.

Summary:

Overall, the reviewers found this to be an interesting study on another aspect of cerebellar organization into modules. Using several molecular markers for the inhibitory interneuron basket cell pinceau as well as a genetic tracing approach, the data nicely demonstrate that basket cell pinceau are organized into different zones defined by size of the basket cell pinceau onto Purkinje cell soma and initial segments.

Essential Revisions:

1) Impact of this study could be enhanced if there were more insight/information into what pathways/processes influence pinceau size. For example, are pinceau zones impacted in the absence of basket cell activity? Another point would for the authors to enhance the discussion on how basket cell pinceau size is impacted by decreasing Purkinje cell neurotransmission. What are potential dots connecting Purkinje cell transmitter release with basket cell pinceau size?

2) Quantification is often lacking. Example images are often not convincing. Details of analysis in the Materials and methods section suggests that blinding and other approaches to avoid bias (e.g. in selecting particular ROIs) were not used.

3) How did the analysis in *Pcp2-Slc32a1* mutants circumvent the confound of altered zone boundaries? Is it that pinceau size regulation is lost, or that it can no longer be easily observed without clear Purkinje cell bands?

[Editors' note: further revisions were suggested prior to acceptance, as described below.]

Thank you for resubmitting your work entitled "Purkinje cell neurotransmission patterns cerebellar basket cells into zonal modules defined by distinct pinceau sizes" for further consideration by *eLife*. Your revised article has been evaluated by K VijayRaghavan (Senior Editor) and a Reviewing Editor.

Summary:

Overall, the reviewers found the revised manuscript to be much improved. This interesting study uses molecular markers for the inhibitory interneuron basket cell pinceau as well as a genetic tracing approach, to very nicely demonstrate that basket cell pinceau are organized into different zones defined by size of the basket cell pinceau onto Purkinje cell soma and initial segments.

While the manuscript is much improved, there are some remaining issues that need to be addressed before acceptance, as outlined below:

1) Please revise wording regarding Purkinje cell "activity", "firing rate", or "electrophysiology" to "neurotransmission", especially as White et al. reported no change in simple spike rate in the mutants.

2) Quantification is lacking for some figures. In particular Figure 4, make it a more quantitative, rather than qualitative, report.

---

## [Author Response]

Essential Revisions:1) Impact of this study could be enhanced if there were more insight/information into what pathways/processes influence pinceau size. For example, are pinceau zones impacted in the absence of basket cell activity? Another point would for the authors to enhance the discussion on how basket cell pinceau size is impacted by decreasing Purkinje cell neurotransmission. What are potential dots connecting Purkinje cell transmitter release with basket cell pinceau size?

These are excellent suggestions. To address these questions, we have tested whether pinceau sizes are impacted by the lack of basket cell activity by quantifying the area occupied by HCN1 expression in regions of interest within zebrinII-positive and zebrinII-negative Purkinje cell zones in 3 animals lacking basket cell GABA neurotransmission and 3 littermate controls. We have found that HCN1 maintains its zonal patterning even in the absence of basket cell neurotransmission. We have added this experiment to the Materials and methods and Results as well as represented our findings in Figure 8E-K. We have also added to the Discussion additional interpretation of our finding that the lack of Purkinje cell neurotransmission (by genetically deleting VGAT) disrupts the patterning of basket cell pinceaux. The revised Discussion text now reads:

“The differences in baseline Purkinje cell firing rate between zebrinII-positive and zebrinII-negative zones (Zhou et al., 2014; Xiao et al., 2014) raised the intriguing possibility that the establishment of zonal patterns as defined by pinceaux size may be driven by the level of Purkinje cell activity. (…) Regardless, there is compelling evidence that Purkinje cell neurotransmission and their in vivo electrophysiological properties have an intriguing influence over basket cell pinceau morphology and size.”

2) Quantification is often lacking. Example images are often not convincing. Details of analysis in the Materials and methods section suggests that blinding and other approaches to avoid bias (e.g. in selecting particular ROIs) were not used.

We appreciate these comments. To address first part of the comment, we have added additional quantification for the data (see above). We have also gone through the entire manuscript to ensure that we have provided high quality images in all panels and we have made specific adjustments to certain images as requested by the individual reviewers. We have also clarified our ROI selection criteria in the Materials and methods section. We now describe that in our analyses, for the quantification of pinceaux size, ROI location was determined based on the border of expression of Purkinje cell zonal markers and was always set to the same size. For quantification of pinceaux fluorescence intensity, the same size ROI was used for all pinceaux quantified. We have added the following text to the Materials and methods section:

“Purkinje cell zonal boundaries within the image were identified by an experimenter trained to examine cerebellar anatomy and cellular architecture. For the purposes of this study, the zonal boundaries were defined as the abutting region(s) that comes directly after the last Purkinje cell that expresses a given zonal marker.”

“This method of defining the ROI, using combined molecular expression and anatomy to identify a zonal boundary paired with analyses that consider a standardized and uniform ROI area across all images, served to limit the potential for bias in our analyses.”

“Pinceaux in both zebrinII-positive and -negative zones were evenly selected across the different zones for fluorescence analysis.”

3) How did the analysis in Pcp2-Slc32a1 mutants circumvent the confound of altered zone boundaries? Is it that pinceau size regulation is lost, or that it can no longer be easily observed without clear Purkinje cell bands?

This is an important point. It is true that we have previously found that *Pcp2^Cre^;Slc32a1^flox/flox^* mice have poorly defined Purkinje cell zebrinII zonal boundaries (White et al., 2014). We have added the following clarification to our Materials and methods section:

“In a previous study, we showed a blurring of Purkinje cell zonal boundaries in the *Pcp2^Cre^;Slc32a1^flox/flox^* mice (White et al., 2014). Here, we analyzed cerebellar zonal properties while accounting for the presence of blurred domains by setting the zonal boundary at the very last Purkinje cell that clearly expressed the zebrinII zonal marker, which established a “blurred” zonal region on one side of the defined boundary and a “pure” zonal region on the other. Defining the zonal patterns in this manner aided the quantification of zonal defects in the mutant mice.”

As zebrinII staining was used as our zonal marker for this analysis and our primary difference was found in zebrinII-negative zones, we could be sure that we were comparing “pure” zebrinII-negative zones between the *Pcp2^Cre^;Slc32a1^flox/flox^* and control mice. As zebrinII expression is not reduced, but extended in its distribution, it still forms an easily observable boundary between this “pure” zebrinII-negative and blurred zone. Therefore, we also added the following text to the Results:

“Despite the blurring of Purkinje cell zonal boundaries in the *Pcp2^Cre^;Slc32a1^flox/flox^* mutant mice (White et al., 2014), we were still able to analyze zonal properties by accounting for the presence of the blurred regions. We set the zonal boundary at the very last Purkinje cell that clearly expressed the zonal marker, which effectively created a “blurred” zonal region on one side of the defined boundary and a “pure” zonal region on the other, which were then used for quantification. Therefore, our data showing reduced pinceau size in the zebrinII-negative zones does not represent a mixing of cellular identities in a particular region that contains the large pinceaux on zebrinII-negative Purkinje cells diluted by the small pinceaux on zebrinII-positive cells, but rather the data suggest that the zebrinII-negative Purkinje cells in the mutant are innervated by pinceau with reduced sizes.”

[Editors' note: further revisions were suggested prior to acceptance, as described below.]

While the manuscript is much improved, there are some remaining issues that need to be addressed before acceptance, as outlined below:1) Please revise wording regarding Purkinje cell "activity", "firing rate", or "electrophysiology" to "neurotransmission", especially as White et al. reported no change in simple spike rate in the mutants.

We agree that our terminology could be further clarified and have made updates to the text throughout the manuscript accordingly. However, please note that in a few specific instances we have left the wording as it was before. In these cases, which are exclusively in relation to citations of previous work by other groups, we left the more general wording such as “activity” because those studies did not distinguish between neurotransmission from other modes of neuronal communication.

2) Quantification is lacking for some figures. In particular Figure 4, make it a more quantitative, rather than qualitative, report.

Thank you for this suggestion. We have performed quantification for Figure 4 and added a new figure panel, Figure 4K, to represent our results. We have updated the figure legend to add relevant information for this new panel and added the following text to the Results section:

“We quantified both the size of the HCN1-expressing pinceau region as well as NFH expression, which is localized to both the pinceau as well as the Purkinje cell (Demilly et al., 2011), in the pinceau region (Figure 4K). We found that the pinceau region revealed by both of these markers was larger in NFH-positive zones compared to the negative zones. Additionally, there was no overlap of pinceau size between positive and negative zones among any of the lobules included in the analysis.”